# BAD-LAMP controls TLR9 trafficking and signalling in human plasmacytoid dendritic cells

Alexis Combes[1], Voahirana Camosseto[1,2], Prudence N'Guessan[1], Rafael J. Argüello[1], Julie Mussard[3,4,5,6], Christophe Caux[3,4,5,6], Nathalie Bendriss-Vermare[3,4,5,6], Philippe Pierre [1,2,7] & Evelina Gatti[1,2,7]

Toll-like receptors (TLR) are essential components of the innate immune system. Several accessory proteins, such as UNC93B1, are required for transport and activation of nucleic acid sensing Toll-like receptors in endosomes. Here, we show that BAD-LAMP (LAMP5) controls TLR9 trafficking to LAMP1[+] late endosomes in human plasmacytoid dendritic cells (pDC), leading to NF-κB activation and TNF production upon DNA detection. An inducible VAMP3[+]/LAMP2[+]/LAMP1[−] endolysosome compartment exists in pDCs from which TLR9 activation triggers type I interferon expression. BAD-LAMP-silencing enhances TLR9 retention in this compartment and consequent downstream signalling events. Conversely, sustained BAD-LAMP expression in pDCs contributes to their lack of type I interferon production after exposure to a TGF-β-positive microenvironment or isolation from human breast tumours. Hence, BAD-LAMP limits interferon expression in pDCs indirectly, by promoting TLR9 sorting to late endosome compartments at steady state and in response to immunomodulatory cues.

[1] Aix Marseille Université, CNRS, INSERM, CIML, 13288 Marseille cedex 9, France. [2] International associated laboratory (LIA) CNRS "Mistra", 13288 Marseille cedex 9, France. [3] Centre Léon Berard, 69373 LYON cedex 08, France. [4] Université de Lyon, 69373 LYON cedex 08, France. [5] INSERM U1052, 69373 LYON cedex 08, France. [6] CNRS UMR5286, 69373 LYON cedex 08, France. [7] Institute for Research in Biomedicine-iBiMED and Aveiro Health Sciences Program University of Aveiro, Aveiro 3810-193, Portugal. Evelina Gatti and Philippe Pierre contributed equally to this work. Correspondence and requests for materials should be addressed to E.G. (email: gatti@ciml.univ-mrs.fr)

Members of the Toll-like receptor (TLR) family are innate immune sensors for pathogen-associated molecular patterns, ranging from bacterial cell wall components to viral and bacterial nucleic acids[1]. TLRs use a set of signalling adaptors to induce a transcriptional response, which leads to pro-inflammatory cytokines and type I interferon (IFN) production[2, 3]. TLRs are divided into two subgroups according to extracellular or intracellular localisation and respective ligands[4]. Intracellular TLRs, such as TLR3, TLR7 and TLR9, recognise nucleic acids and need to be proteolytically activated in endosomes to signal[5–7]. For instance, cleaved TLR9 is activated by unmethylated cytosine-phosphate-guanine dinucleotides (CpG) DNA motifs that are frequently present in microbes but are rare in mammalian cells[8, 9]. Hence, in physiological conditions, TLR9 is a sensor of infection[10]; however, in mice and in patients with lupus-like symptoms, TLR9 can be activated by immune complexes formed with self DNA[11]. Tightly controlled TLR9 transport and activation seem, therefore, to be particularly important to prevent autoimmunity and discriminate self from non-self DNA[12]. Subcellular distribution of TLRs correlates well with membrane compartments, in which ligands are found and can drive activation of different signalling cascades[13]. Endocytic TLRs require an additional accessory protein, the uncoordinated 93 homologue B1 (UNC93B1) to leave the endoplasmic reticulum (ER) and reach endosomes to be activated[14–17]. However, the mechanisms controlling the initiation of UNC93B1–TLR complex transport from the ER to endosomes are not clear. Upon entry of TLRs in the endocytic pathway, additional sorting to specific signalling compartments, from which IRFs or NF-κB can be activated respectively, is required. Regulated access to IRF-signalling endosomes (SE) or NF-κB-SEs, is thus necessary to coordinate IFNs and pro-inflammatory cytokine production[18–20]. Supporting the importance of this sorting step, DCs and macrophages isolated from adaptor protein 3 (AP-3)-deficient mice[21], or from patients with Hermansky–Pudlak syndrome type 2 (HPS2)[22], are deficient in late endosomal transport and consecutive type I IFN production. However, these deficiencies do not, or only moderately, affect pro-inflammatory cytokines expression in response to TLR9 stimulation[21].

pDCs have a unique capacity to produce a large amount of IFN in response to nucleic acids[3] and display specialized molecular features that control IRF7 activation directly downstream of TLR7 and TLR9. In addition to granulin[23], which favors CpG ODN capture, other molecules, including the solute carrier protein superfamily member Slc15a4 or the biogenesis of lysosome-related organelles complexes (BLOC) proteins are required by pDCs, but not by conventional DCs to respond to DNA[24, 25]. pDCs seem, therefore, to have developed a specific regulation of their endocytic compartments, that controls TLR access to different endosome subsets to achieve a coordinated and commensurate DNA detection response to potential threats.

The brain and DC-associated LAMP-like molecule (BAD-LAMP/LAMP5), which shares sequence and structural homology with the canonical lysosomal-associated membrane proteins LAMP1 and LAMP2, is expressed in the nervous tissues of most metazoan species[26, 27]. BAD-LAMP is also expressed by non-activated human pDCs and blastic pDCs neoplasms (BPDCN) from leukaemic patients, enabling rapid identification in tissues and blood[28].

Here, we show using human primary pDCs and the BPDCN-derived CAL-1 cell line that BAD-LAMP controls the sorting of TLR9 in different endosome subsets and favors pro-inflammatory cytokine production. Upon CpG detection, BAD-LAMP is transported with TLR9 to the IRF7-SE, where it promotes further TLR9 sorting to LAMP1+ NF-κB-SE. Inhibition of BAD-LAMP activity therefore promotes TLR9 retention in IRF7-SE and leads

to increased IFN expression. Confocal microscopy analysis of IRF7-SE shows that this SE is a CpG-inducible endosomal hybrid/intermediate compartment containing both the sorting endosomes-associated SNARE protein vesicle-associated membrane protein 3 (VAMP3) and LAMP2, but not its close relative LAMP1. We further show that pDCs exposure to immunosuppressive cytokines or tumour supernatants prevents the down-modulation of BAD-LAMP, which is normally rapid after activation by CpG, and consequently limits type I IFN production. The sustained BAD-LAMP expression in breast tumour-associated pDCs is therefore likely to contribute to the lack of IFN-α production, a dysfunctional phenotype generally associated with immune tolerance and aggressive cancer[29, 30].

## Results

**BAD-LAMP is downregulated by type I IFN in human pDC.** Previous studies established that the BPDCN-derived CAL-1 cells share many of phenotypic and functional properties of freshly isolated human CD123 + /BDCA4 + pDCs[31, 32]. BAD-LAMP expression was monitored both in isolated primary pDCs from blood and CAL-1 cells using intracellular flow cytometry (Fig. 1a). BAD-LAMP, TLR9 and UNC93B1 were found expressed at high levels in both cell types. After 24 h of CpG-A activation a strong decrease in both BAD-LAMP and TLR9 expression was observed (Fig. 1a). BAD-LAMP disappearance was associated with a reduction in mRNA levels (Fig. 1b), initiated 3 h after CpG stimulation and concomitantly to the type-I IFN production peak observed in both cell types (Fig. 1c). On the basis of the IFN-α and TNF mRNA expression, we could recapitulate the existence of a bimodal activation of TLR9 signalling by CpG-A with first an IRF7-dependent phase (red), followed by an NF-κB-dependent phase (blue; Figs. 1c, 2)[33]. Examination of the signalling pathways downstream of TLR9 by immunoblots (Fig. 1d) confirmed that in CAL-1, the TBK1/IRF7 cascade leading to type-I IFN transcription, was activated by CpG-A within the first hours of treatment, while the peak of NF-κB (p65) phosphorylation appeared after 3 h of treatment. Interestingly, BAD-LAMP down-modulation was observed upon stimulation with all types of CpG ODN[28]. However, with CpG-A, it was initiated at the transition step between the two signalling phases (Fig. 1b, d, e), concomitantly with IFN receptor (IFNAR) signalling, monitored by STAT1 phosphorylation (Fig. 1d). Exogenous IFN-α treatment and IFNAR inhibition with antagonistic antibodies further demonstrated that BAD-LAMP disappearance is caused by IFNAR stimulation (Supplementary Fig. 1). An observation confirmed by using the cysteine protease inhibitor ZA-FA-FMK that prevents TLR9 processing[7] and consequently IFN-α induction (Supplementary Fig. 2A). Upon CpG stimulation, BAD-LAMP is rapidly downregulated both at mRNA and protein levels[28] (Fig. 1c–e). While IFNAR signalling was strongly correlated with transcriptional shut-down (Supplementary Fig. 1B), we further investigated whether BAD-LAMP proteolysis was due to lysosomal targeting as expected from a bona-fide LAMP family member. Both Bafilomycin A1 and Chloroquine, two drugs neutralising endosomal pH, prevented BAD-LAMP protein degradation (Supplementary Fig. 2B, C) and promoted its accumulation in Lamp1 + compartment (Supplementary Fig. 2D), confirming that BAD-LAMP is normally targeted and rapidly degraded in late endosomes/lysosomes. However, upon TLR9 or IFNAR triggering, BAD-LAMP mRNA is rapidly downregulated causing the protein disappearance within few hours of pDC stimulation.

**BAD-LAMP co-localises with UNC93B1 and TLR9 during activation.** BAD-LAMP downregulation upon TLR9 activation

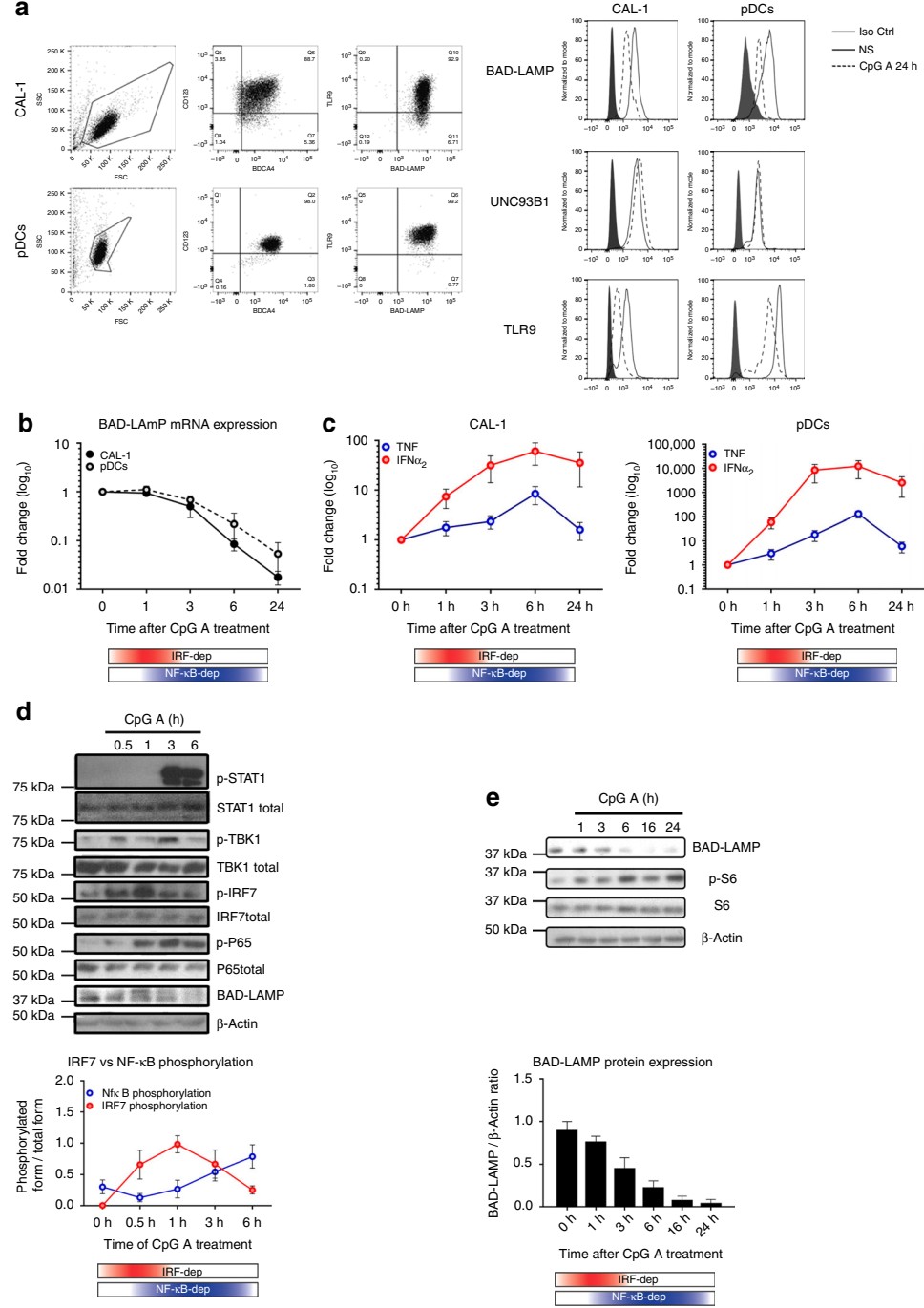

**Fig. 1** BAD-LAMP is down-modulated after IRF7 activation. **a** (*left*) Flow cytometry analysis of CAL-1 cells (*top*) and freshly isolated pDCs (*bottom*) from healthy donors stained for both extracellular (CD123 and BADCA4) and intracellular (TLR9 and BAD-LAMP) pDCs markers. (*right*) Flow cytometry histogram plots of intracellular staining for BAD-LAMP, TLR9 and UNC93B1 in both CAL-1 (*left*) and freshly isolated pDCs (*right*) at steady state (*black line*) and after 24 h of CpG-A stimulation (*dashed line*). Full *grey* histograms represent isotype controls staining. Data are representative of a minimum of three independent experiments. **b** CAL-1 (*black line*) and freshly isolated pDCs (*dashed line*) were treated with CpG-A for indicated times. BAD-LAMP mRNA levels were measured by RT-qPCR. Raw data have been normalised to housekeeping gene (GAPDH) and graphics represent fold change $\pm$ s.d. compared to non-stimulated cells from a minimum of three independent experiments. **c** IFN$\alpha_2$ (*red*) and TNF (*blue*) mRNA level from CAL-1 (*left*) and freshly isolated pDCs (*right*) were measured by RT-qPCR. Raw data have been normalised to housekeeping gene (GAPDH) and graphics represent fold change $\pm$ s.d. compared to non-stimulated cells from a minimum of three independent experiments. **d** (*top*) CAL-1 cells were treated with CpG-A for indicated times prior lysis and sodium dodecyl sulphate–polyacrylamide gel electrophoresis treatment. Expression of BAD-LAMP, TBK1, IRF7, STAT1, the p65 NF-κB subunit, Ribosomal S6 and their phosphorylated forms were detected by immunoblot. β-actin is shown as loading control. **d** (*bottom*) Quantification of IRF7 (*red*) and p65 (*blue*) phosphorylation levels. Graphics represents data normalised to total form levels for each protein $\pm$ s.d. from three independent experiments. **e** (*bottom*) Quantification of BAD-LAMP protein expression by Image J pixel quantification. Graphics represents data normalised β-actin levels $\pm$ s.d. from three independent experiments. The bimodal regulation of TLR9 signalling is represented with a shading *red rectangle* for its IRF7-dependent phase and *blue* for its NF-κB-dependent phase

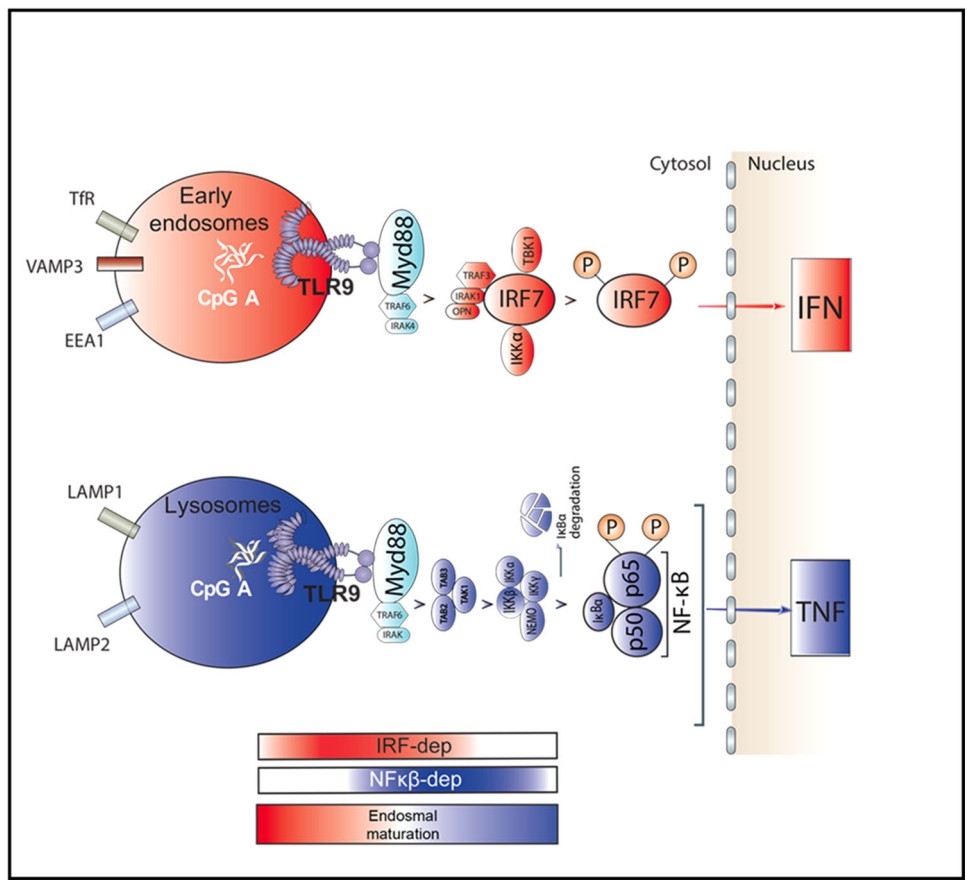

**Fig. 2** Graphical abstract of TLR9 trafficking and signalling in human pDCs. **a** Graphical abstract summarizing the results shown in Fig. 1 on the spatiotemporal organisation of endosomal TLR9 signalling in human pDCs. The bimodal regulation of TLR9 signalling as well as endosomal maturation is represented using rectangles in shading *red* for IRF7-dependent phase and early endosome and in shading *blue* for NF-κB-dependent phase and late endosomes

lead us to investigate BAD-LAMP, TLR9 and its signalling adaptor MYD88 subcellular distribution by performing multicolour confocal microscopy on both freshly isolated primary blood pDC and CAL-1 at key time points of activation by CpG-A. At steady state, BAD-LAMP, TLR9 and MYD88 were poorly co-distributed (Fig. 3a). After 1 h of stimulation, the degree of co-localisation between the three proteins became maximal (*white arrows*; Pearson's coefficient Fig. 3a right). With time, BAD-LAMP and TLR9 co-localisation steadily decreased, until BAD-LAMP down-regulation was fully effective (6 h). As expected from IRF7 phosphorylation and cytokines expression levels (Fig. 1), TLR9 and MYD88 co-localisation reached its peak at 1 h after CpG stimulation and was maintained for at least 5 h (Fig. 3a, Pearson's coefficient). The antibody used to detect endogenous TLR9 seems incapable to recognise the pre-Golgi form of TLR9 and mostly allows visualisation of functional TLR9 in endosomes[6], thus potentially impairing our capacity to reveal any pre-existing interactions with ER-resident TLR9 prior processing and pDC activation. Nevertheless, the co-distribution of BAD-LAMP with processed TLR9 and MYD88 after 1 and 3 h of activation, was confirmed using voxel gating imaging to reveal areas of triple co-localisation with BAD-LAMP (Fig. 3b), suggesting that the three molecules reside in close vicinity in the same endocytic compartment at a key moment of TLR9 signalling.

We increased the resolution of our analysis by performing confocal imaging using in situ Proximity Ligation Assay (iPLA)[34] to quantify the proximity of BAD-LAMP with UNC93B1 and

TLR9 (Figs. 3c d). As anticipated from BAD-LAMP and UNC93B1 co-localisation in HeLa cells[28], BAD-LAMP and UNC93B1 were found in close proximity (≤40 nm) in non-activated pDCs. This proximity was rapidly lost upon CpG stimulation and replaced by a strong co-localisation between BAD-LAMP and TLR9, observed concomitantly with MYD88 recruitment after 1 h of activation (Figs 3d e). However, TLR9 vicinity with BAD-LAMP faded with time, while its association with MYD88 persisted (Fig. 3e). These observations suggest that BAD-LAMP and UNC93B1 co-exist in the ER of human pDC at steady state, and that upon activation, this interaction is substituted by a novel and exclusive relationship between BAD-LAMP and functional TLR9 molecules in signalling endosomes. A closer examination of iPLA imaging in non-activated pDCs revealed a weak but nevertheless pre-existing interaction of TLR9 with MYD88 (Fig. 3e), that is difficult to visualise by conventional confocal microscopy, given the relatively broad and overlapping intracellular distribution of these molecules. A small fraction of processed endosomal TLR9 is therefore readily available to detect DNA, explaining how the response to CpG ODN and a wave UNC93B1-dependent TLR9 recruitment could be initiated[16, 35].

**TLR9 is targeted to an inducible VAMP3+LAMP2+ compartment.** We demonstrated that in the first hour of pDCs activation, BAD-LAMP loses its vicinity with UNC93B1 and associates or

remains associated with TLR9 in endosomes. We further dissected BAD-LAMP and TLR9 intracellular transport by following several markers of known endosome subsets. VAMP-3 is generally associated with recycling and sorting endosomes, since it facilitates the transit of recycling receptors (e.g., Transferrin receptor)[36]. VAMP3 distribution was compared to LAMP1 one that mostly marks late endosomes and lysosomes, no

significant overlap between the two molecules was observed at any time during pDC activation (Fig. 4a and Supplementary Fig. 3A). Unexpectedly, LAMP2, which is normally found tightly associated in late endosomes with LAMP1, was found to segregate from LAMP1-positive compartments after 1 h of CpG activation and to acquire a common localisation with VAMP3 (Fig. 4a *white arrows*, Pearson's coefficient right and voxel gating in

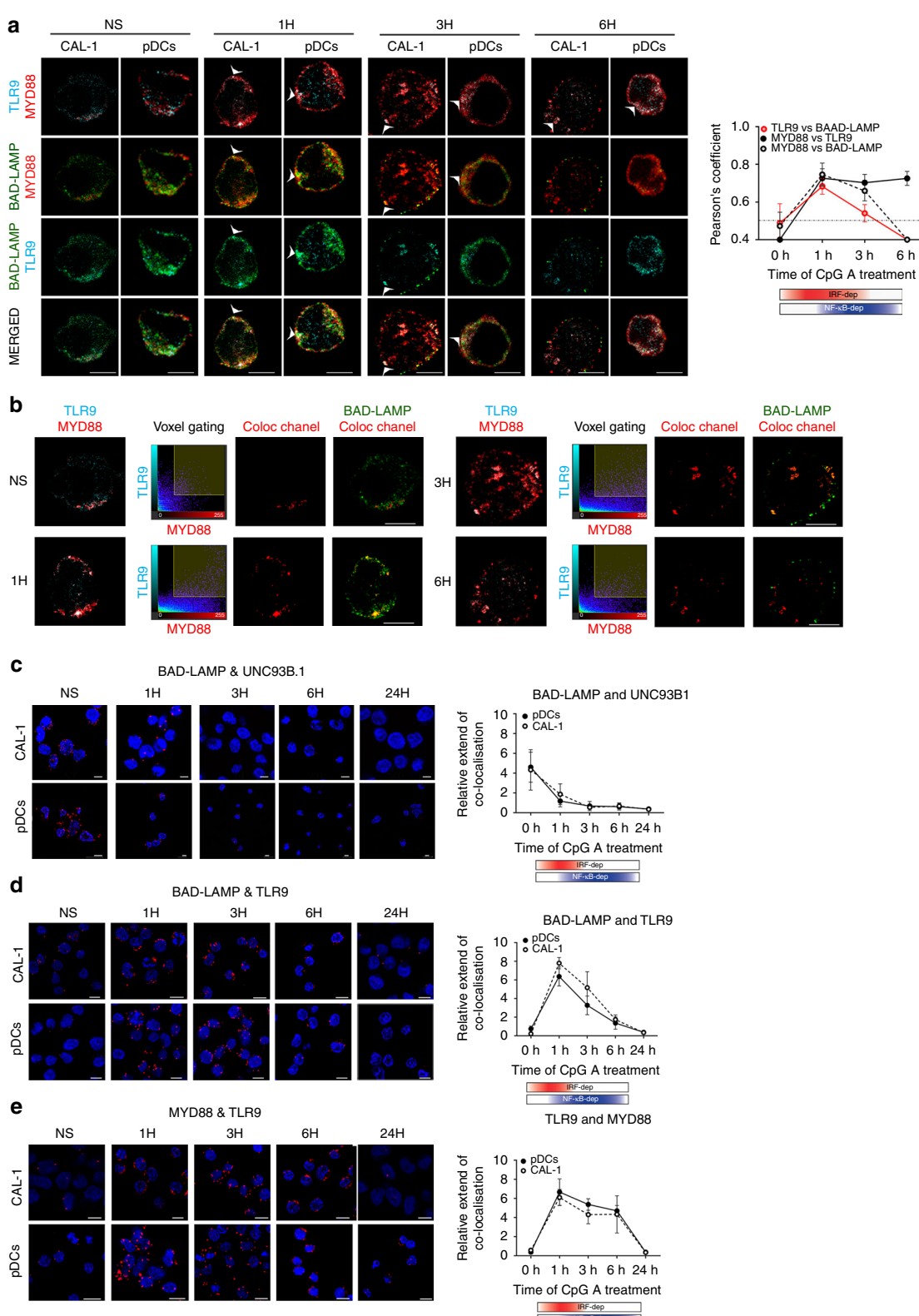

Supplementary Fig. 3B). This dynamic process peaked at 3 h of CpG activation, before returning to a steady-state distribution with LAMP2 again mostly overlapping with LAMP1, away from VAMP3 (Fig. 3a). CpG ODN stimulation induces therefore a rapid reorganisation of endosomes in pDC, during which VAMP-3 and LAMP2 are co-distributed in a hybrid compartment distinct from classical late LAMP1+ compartments. 3D reconstruction of several sets of images (Supplementary Fig. 3A), further indicated that LAMP1 and LAMP2 are separated in distinct domains of the same late endosomes segregated from VAMP3+ organelles. Upon the first hours of pDC activation, partitioning of LAMP1 and LAMP2 was increased, while LAMP2 and VAMP3 initiated to merge in one or several continuous compartments, independently of LAMP1 distribution (Supplementary Fig. 3A and Fig. 5a). Finally, the endosomal system returned to equilibrium after 6 h of CpG ODN stimulation with the restoration of LAMP1/LAMP2 co-distribution and disappearance of the mixed compartment containing both VAMP3 and LAMP2 (Supplementary Fig. 3A).

Upon activation, TLR9 was found to access immediately VAMP3+ compartments prior exiting from these endosomes and entering in late LAMP1+ compartments 3 to 6 h after CpG exposure (Fig. 4b white arrows, and Supplementary Fig. 4A). Interestingly MYD88 distribution followed closely TLR9 trafficking, co-localising with VAMP3 and LAMP2 in the early phase of activation (Fig. 4c) and augmenting its distribution within classical LAMP1/LAMP2 double-positive late endosomes later on (Fig. 4c). IRF7 recruitment in the hybrid compartment was confirmed by voxel gating imaging showing that TLR9, IRF7 co-localise with VAMP-3+ and LAMP2+ upon 1-3 h of CpG stimulation, suggesting that CpG activation primarily drives TLR9/MYD88 association in VAMP-3+/LAMP2+ organelles, from which IRF7 activation is initiated (Supplementary Fig. 3B). After 6 h of activation, concomitantly with LAMP2 redistribution, TLR9 and MYD88 were found in late LAMP1+ endosomes from which they likely trigger NF-κB activation (Fig. 4c and Supplementary Fig. 3B). During this process, BAD-LAMP, together with TLR9 and MYD88, reached VAMP3+ endosomes at early time points (Fig. 4d and Supplementary Fig. 4A), prior disappearing gradually during time (Fig. 4d). Given the dynamic nature of VAMP3+ endosomes, we further defined their molecular identity by using Syntaxin 6 (STX6), which is a SNARE associated with post-TGN endosomes. VAMP3 sub-distributed in two types of organelles, in a pericentriolar compartment positive for STX6, and alternatively in a population of peripheral endosomes negative for STX6 (Supplementary Fig. 4A). Upon CpG activation, BAD-LAMP and TLR9 were found to accumulate exclusively in VAMP3+ peripheral endosomes, while pericentriolar STX6+/VAMP3+ organelles remained devoid of staining (Supplementary Fig. 4A),

in accordance with their presumed recycling endosomes identity[37].

Since CpG-A ODN traffic through different signalling compartments sets the pace for pDCs activation, we imaged soluble fluorescent CpG-A distribution after uptake by primary pDCs and CAL-1 over time (Supplementary Fig. 4C, D). CpG-A followed the same intracellular path previously observed for TLR9 and MYD88, being first in direct vicinity of TLR9 and BAD-LAMP in VAMP3+/LAMP2+ compartment, and then accessing at later time the LAMP1+ late endosomes (Supplementary Fig. 4C, D white arrows). Thus, in human pDCs, TLR9 activation by CpG-A is spatiotemporally organised by first triggering type-I IFN expression in an inducible VAMP3+/LAMP2+ chimeric endosomal compartment, prior initiating pro-inflammatory cytokines production from conventional LAMP1+ late endocytic compartments (Fig. 5a). This situation is strikingly converse to what has been previously observed in mouse macrophages[21, 38], but in line with results dissecting CpG-A activity on mouse DC[39].

**BAD-LAMP inhibits type I IFN induction by CpG ODN**. We next silenced BAD-LAMP expression in CAL-1 cells to address its possible role on TLR9 trafficking and signalling. RNA interference was performed using as positive control, UNC93B1 siRNA, which should abolish the pDC response to CpG ODN. Upon siRNA introduction, BAD-LAMP expression was efficiently downregulated both at the mRNA and protein level (Supplementary Fig. 5A), while scramble siRNA did not interfere with BAD-LAMP expression at steady state, nor with its down-modulation upon stimulation. Conversely, UNC93B1 silencing by blocking TLR9 recruitment to endosomes (Fig. 6a) prevented pDC activation and associated BAD-LAMP decay (Supplementary Fig. 5A). Importantly, BAD-LAMP silencing strongly affected TLR9 trafficking and promoted its accumulation in VAMP3+ compartments, while sorting to LAMP1+ late endosomes was inhibited over time (Fig. 6a). This was particularly obvious in non-stimulated cells in which BAD-LAMP silencing induced a strong co-localisation of TLR9 with VAMP3, equivalent to what is observed after 3 h of CpG activation (Fig. 6a) and completely absent from cells treated with scramble siRNA. Mirroring its accumulation in VAMP3 + compartments, TLR9 co-localisation with LAMP1 was considerably decreased upon BAD-LAMP extinction (Fig. 6a). All these observations were confirmed by performing iPLA on RNAi-treated cells (Supplementary Fig. 5B) in which BAD-LAMP silencing induced a strong and persistent accumulation of TLR9 in close vicinity of VAMP3 and promoted IRF7 recruitment to this compartment (Supplementary Fig. 5C).

We investigated how enhanced accumulation of TLR9 and IRF7 in VAMP3+ peripheral endosomes impacted signalling in

**Fig. 3** BAD-LAMP is co-localised with TLR9 and MYD88 upon activation by CpG. **a** (left) Immunofluorescence confocal microscopy (ICM) of CAL-1 and freshly isolated pDCs stimulated or not (NS) with CpG-A for indicated times was performed to visualise BAD-LAMP (green), MYD88 (red) and TLR9 (cyan). Arrowheads point at co-localisation area. (right) Co-localisation quantifications using Pearson's coefficient measurement (ImageJ) are shown for TLR9 and BAD-LAMP (red line), MYD88 and TLR9 (black line), MYD88 and BAD-LAMP (dashed line) during time. Graphics represent Pearson's coefficient means of 50 different cells ± s.d. for each time point from at least three independent experiments. **b** Voxel gating analysis of immunofluorescence confocal images of CAL-1 cells presented in **a**. Voxel gating was performed on TLR9 (cyan) and MYD88 (red) channels to generate a 'coloc channel' picture only showing co-localisation areas between the two proteins. This 'coloc channel' was then merged with single BAD-LAMP staining to obtain simplified tri-localisation images, revealing strong BAD-LAMP co-localisation with TLR9/MYD88 after 1 h of CpG-A exposure. Scale bars = 5 μM. **c–e** Immunofluorescence proximity ligation assay (iPLA) on CAL-1 or freshly isolated pDCs stimulated with CpG-A for indicated times performed for BAD-LAMP and UNC93B1 (**c**), BAD-LAMP and TLR9 (**d**), and TLR9 and MYD88 (**e**). (left) Confocal images are representative of at least three independent experiments in which the nucleus is stained by DAPI (blue) and proximity between the two proteins of interest is revealed by incorporation of labelled nucleotide (red) during the ligation reaction. Scale bars = 10 μM. (right) Quantification of iPLA for pDCs (black line) and CAL-1 (dashed line) by counting the number of red dots normalised to nucleus numbers using imageJ. Graphics are representing means of dots/cell for 150 different cells ± s.d. from at least three independent experiments. The bimodal regulation of TLR9 signalling is represented with a shading red rectangle for its IRF7-dependent phase and blue for its NF-κB-dependent phase

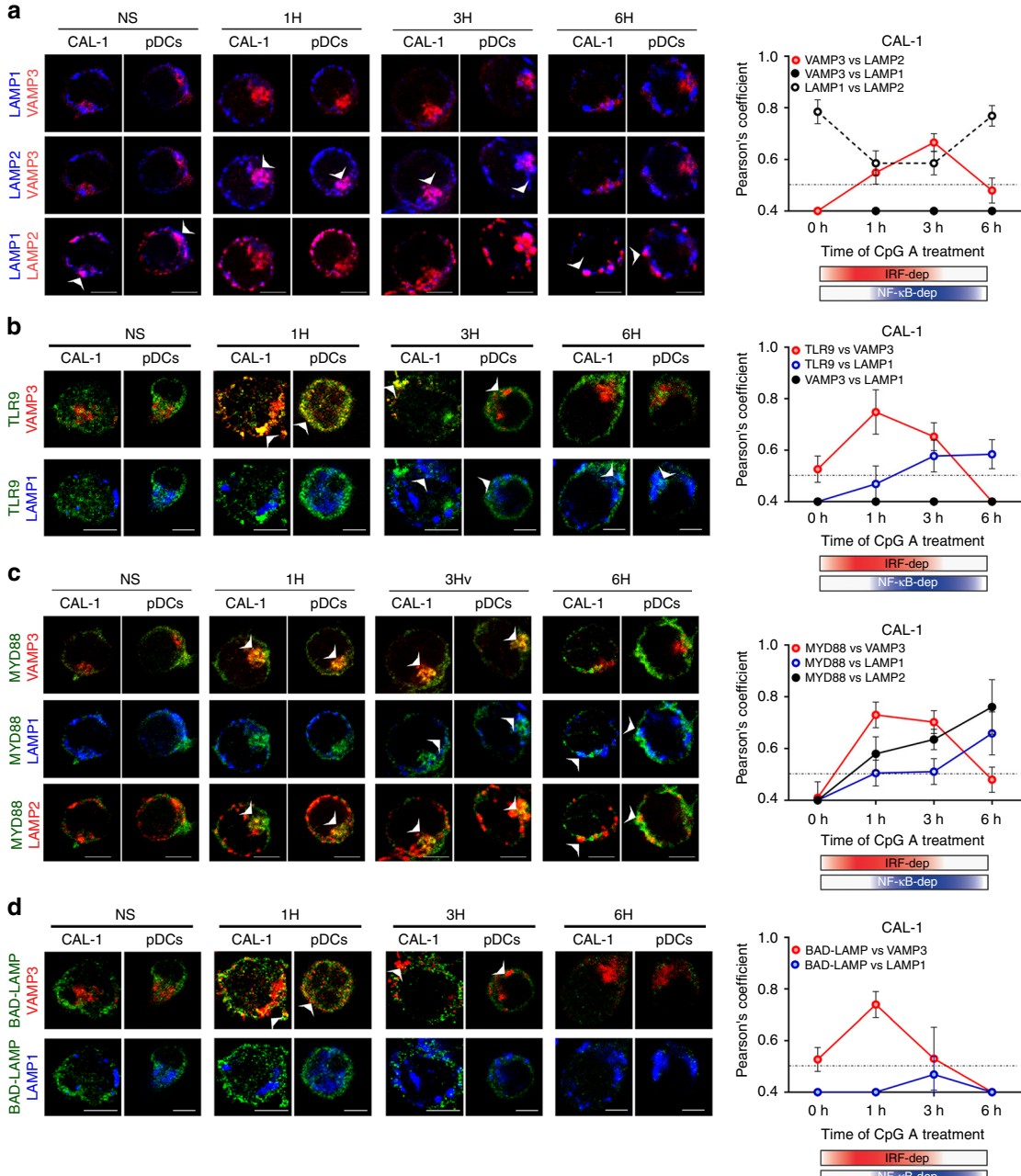

**Fig. 4** Upon activation TLR9 and BAD-LAMP are addressed to specialized endosomes. **a–d** (*left*) Immunofluorescence confocal microscopy (ICM) on CAL-1 or freshly isolated pDCs stimulated with CpG-A for indicated times and stained of (**a**) LAMP1, LAMP2 and VAMP3; (**b**) TLR9, VAMP3 and LAMP1; (**c**) MYD88, LAMP1, LAMP2 and VAMP3; (**d**) BAD-LAMP, VAMP3 and LAMP1. Pictures are representative of at least three independent experiments. *Arrows* indicate co-localisation areas. *Scale bars* = 5 μM. (*right*) **a–d** Quantification of the co-localisation between the different proteins across time was performed by Pearson's coefficient measurement using ImageJ. Graphics represent Pearson's coefficient means of 50 different cells ± s.d. from at least three independent experiments, significance of pixel correlation is only considered above a PCM of 0.5. The bimodal regulation of TLR9 signalling is represented with a shading *red rectangle* for its IRF7-dependent phase and *blue* for its NF-κB-dependent phase

silenced cells by monitoring INF-α2 and TNFα mRNA expression after CpG stimulation (Fig. 6b). As expected, UNC93B1 silencing prevented the induction of both cytokines, while the scramble control had little effect on the bimodal induction of INFα2 and TNFα by CpG (Fig. 6b). Importantly, BAD-LAMP silencing induced INFα2 mRNA expression in absence of any CpG stimulation (8 folds over control) (Fig. 6b), correlating with the strong accumulation of TLR9 in VAMP3+ endosomes at steady state (Fig. 6a). This enhancement was confirmed after CpG treatment, since a 10-fold increase over control in INFα2 mRNA

levels was observed in BAD-LAMP-silenced cells after only 1 h of stimulation, while type-I IFN secretion was augmented threefold after 6 h (Fig. 6c). Increased IFN levels were inversely correlated with TNF expression (Figs. 4c and 6b), which was inhibited by BAD-LAMP silencing, both at the transcriptional and protein secretion levels. Importantly BAD-LAMP disappearance did not impair CpG internalisation (Supplementary Fig. 5D), nor LAMP2 redistribution with VAMP3 (Supplementary Fig. 5E), suggesting that BAD-LAMP regulates the temporal organisation of different signals transduced upon CpG ODN detection mostly

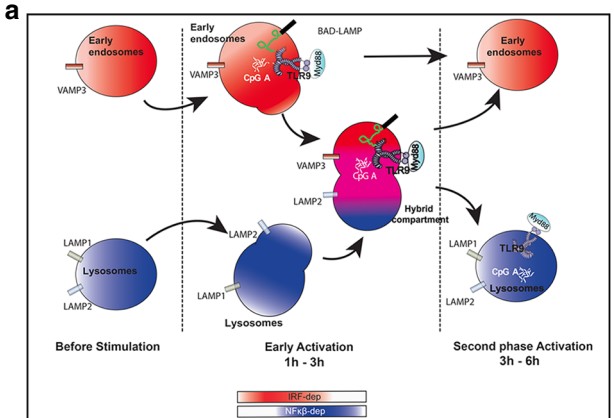

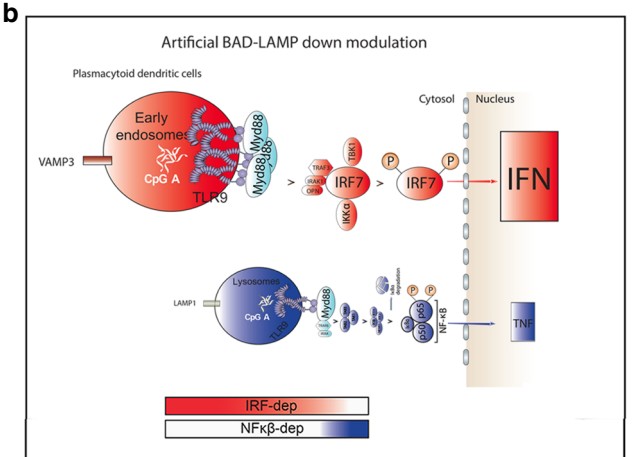

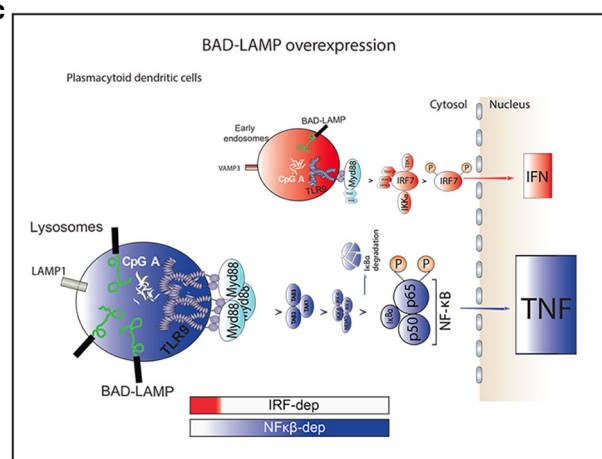

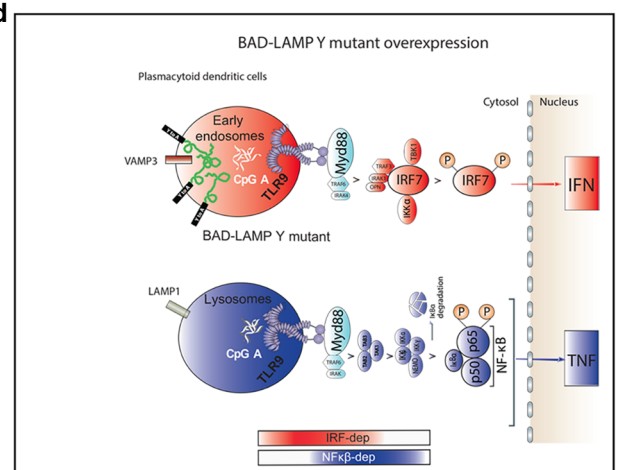

**Fig. 5** Graphical abstracts of TLR9 trafficking and signalling in human pDCs upon activation or BAD-LAMP manipulation. **a** Graphical abstract summarizing the results shown in Fig. 4 on the spatiotemporal organisation of endosomes and TLR9 signalling in CpG-activated human pDCs. **b** Graphical abstract summarizing the results shown in Fig. 6 on the spatiotemporal organisation of endosomes and TLR9 signalling upon BAD-LAMP silencing. **c** Graphical abstract summarising the results shown in Fig. 7 on the spatiotemporal organisation of endosomes and TLR9 signalling upon overexpression of BAD-LAMP. **d** Graphical abstract summarising the results shown in Fig. 9 on the spatiotemporal organisation of endosomes and TLR9 signalling, upon overexpression of BAD-LAMP bearing a mutation in its YxxΦ addressing signal. The bimodal regulation of TLR9 signalling as well as endosomal maturation is represented using rectangles in *shading red* for IRF7-dependent phase and early endosome and in *shading blue* for NF-κB-dependent phase and late endosomes

by interfering with TLR9 sorting and residency in VAMP3⁺ endosomes (Fig. 5b). The phosphorylation/activation kinetics of TBK1, IRF7, AKT and NF-κB (p65) in silenced CAL-1 cells confirmed this hypothesis (Fig. 6d and Supplementary Fig. 5E), since in absence of BAD-LAMP, TBK1 (MFI) and IRF7

phosphorylation was already increased prior by CpG stimulation and maintained for at least 6 h, while P-p65 levels were never increased compared to scramble control. Interestingly, AKT activation was also reinforced in BAD-LAMP-silenced cells, suggesting that AKT activation by TLR9 is also triggered from

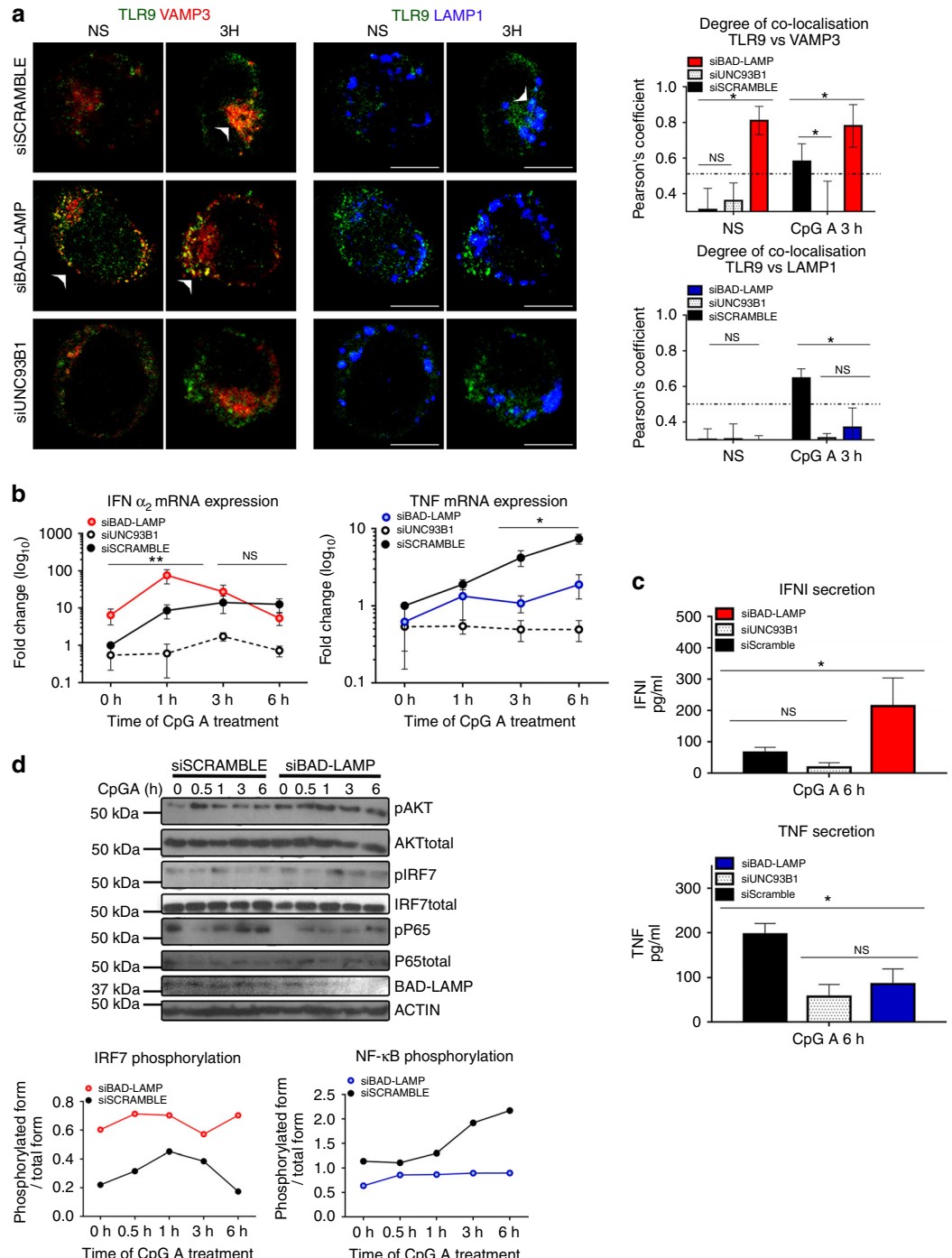

**Fig. 6** BAD-LAMP silencing promotes TLR9-dependent type-I IFN expression. **a**. (*left*) Immunofluorescence confocal microscopy (ICM) on CAL-1 electroporated with the different indicated siRNA and stimulated with CpG-A for indicated times. Analyse at steady state (NS) or 3 h after CpG-A stimulation of LAMP1, VAMP3 and TLR9 distribution. Images are representative of at least three independent experiments. *Arrowheads* indicate co-localisation areas. *Scale bars* = 5 μM. (*right*) Quantification of co-localisation between TLR9 with VAMP3 (*top*), or with LAMP1 (*bottom*) at steady state (NS) or 3 h after CpG-A stimulation, was performed by Pearson's coefficient measurement using ImageJ. Graphic represent Pearson's coefficient means of 50 different cells ± s.d. from at least three independent experiments. **b** IFNα₂ (*left*) and TNF (*right*) mRNA level were monitored by RT-QPCR. Raw data have been normalised to housekeeping gene (GAPDH) and graphics represent fold change ± s.d. compared to non-stimulated cells from three independent experiments minimum. **c** IFNα (*top*) and TNF (*bottom*) secretion were monitored by ELISA at 6 h after CpG-A stimulation. Graphics represent cytokines concentration ± s.d. from three independent experiments minimum. **d** (*left*) CAL-1 cells electroporated with siRNA scramble or siRNA BAD-LAMP were treated with CpG-A for indicated times prior lysis and SDS PAGE treatment. Expression of BAD-LAMP, AKT, IRF7, p65 NF-κB subunit and their phosphorylated forms were detected by immunoblot. β-actin is shown as loading control. (*right*) Quantification of IRF7 (*top*) and p65 (*bottom*) phosphorylation levels normalised to their total form by ImageJ quantification. *P < 0.05 by unpaired student's *t*-test

VAMP3-positive endosomes, in agreement with previous observations demonstrating that AKT activation occurs within 20 min of pDC stimulation by CpG or Flu[40].

Given the tight control on BAD-LAMP expression exerted by TLR9 activation, we evaluated the consequences of increasing BAD-LAMP levels in CAL-1 cells. After electroporation of BAD-LAMP mRNA, protein levels were maintained for at least 6 h after CpG treatment (Supplementary Fig. 6A). Overexpression strongly increased BAD-LAMP subcellular co-localisation with LAMP1 both at steady state and upon activation (Fig. 7a),

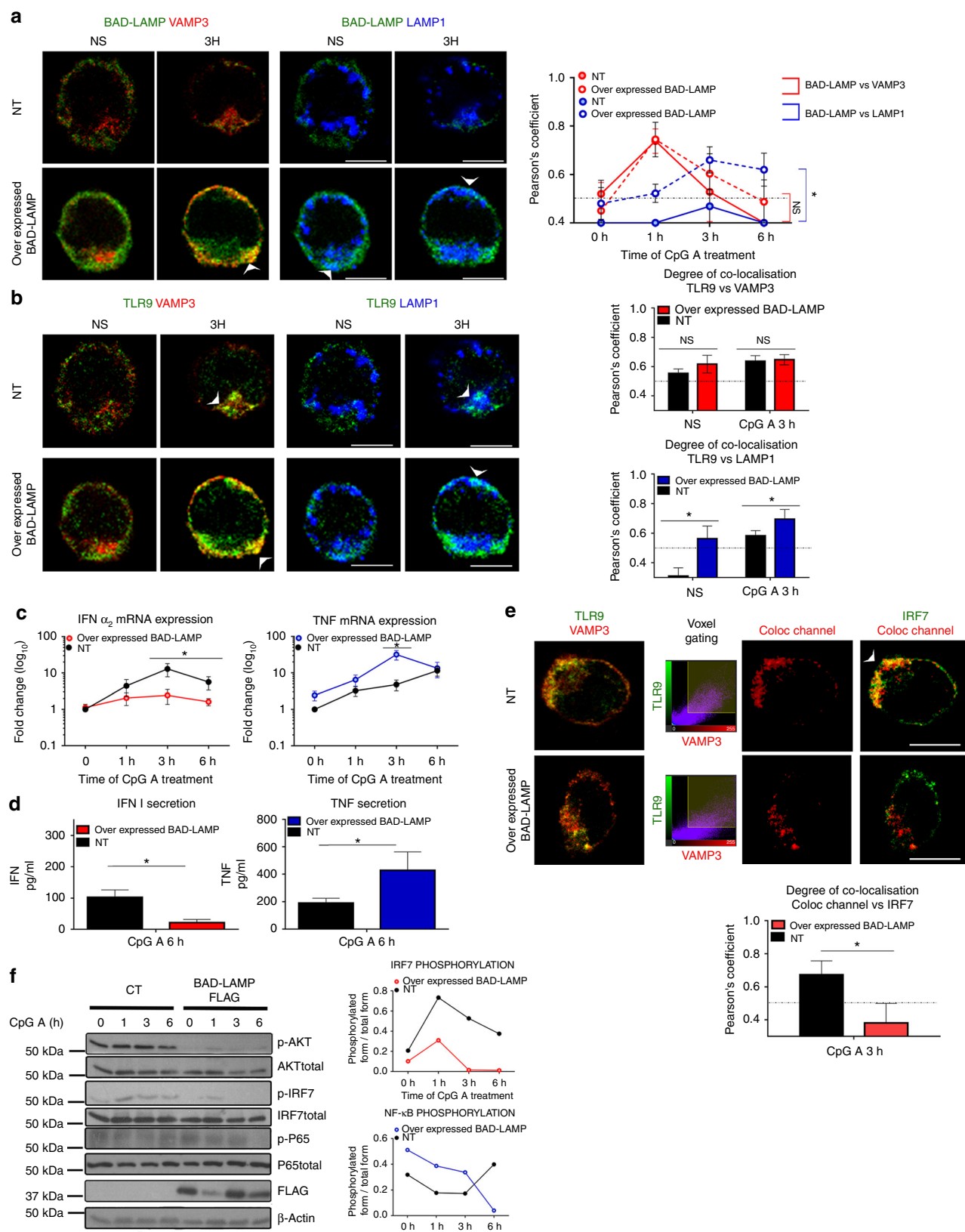

reminiscent of its distribution in Bafilomycin A1-treated cells (Supplementary Fig. 2D). TLR9 transport to VAMP3+ compartment was not affected by BAD-LAMP expression; however, its accumulation in LAMP1+ endosome was strongly increased upon stimulation (Fig. 7b). Enhanced TLR9 targeting to late endosome was accompanied by a reduction in both INF-α2 mRNA expression and secretion (Figs 6d and 7c), while TNF production augmented (Figs 6d and 7c). Although, TLR9 could still be detected in VAMP3+ endosome, its ability to signal was impaired by BAD-LAMP overexpression, which prevented IRF7 recruitment to these TLR9-containing compartments (Fig. 7e, voxel gating). Inhibition of IRF7 recruitment by TLR9 on VAMP3+ endosomes was even more obvious when iPLA was performed on BAD-LAMP-overexpressing cells (Supplementary Fig. 6B, C). In line with TLR9 and IRF7 subcellular localisation, phosphorylation of IRF7, AKT (Fig. 7f) and TBK1 (MFI) (Supplementary Fig. 6E) in response to CpG-A was impaired by BAD-LAMP expression. Conversely, NF-κB (p65) phosphorylation was enhanced, until 6 h of stimulation, a time at which it disappeared. Thus, although BAD-LAMP prevents IRF7 recruitment and accelerates TLR9 access to late endosomes (Fig. 5c), the resulting increase in NF-κB signalling is nevertheless negatively controlled with time, explaining the transcription plateau observed for TNF mRNA levels in Fig. 6c.

Again, these differences in TLR9 transport and associated signalling triggered by BAD-LAMP ectopic expression, could not be attributed to an indirect effect on membrane dynamic or reorganisation, since organelles distribution was normal and CpG-dependent induction of VAMP-3+/LAMP2+ compartments remained unaffected by these treatments (Supplementary Fig. 6D). Taken together both RNA silencing and overexpression experiments, suggest a role for BAD-LAMP in regulating TLR9 signalling by preventing IRF7 recruitment and facilitating sorting from VAMP3+ endosomes to LAMP1 lysosomes (Fig. 5b, c).

**AP-3 and BAD-LAMP YxxΦ motif control TLR9 transport.** Earlier work, monitoring TLR9-GFP ectopic expression in adaptor protein 3 (AP-3) −/− mouse macrophages, has pointed towards a bifurcation of TLR9 transport, which, when targeted by AP-3 to LAMP2+ lysosome-related organelles (LRO), recruits IRF7 and induces type-I IFN[21]. Whether AP-3 is also required for TNF production remains controversial[24], since later work proposed that the carboxyl-terminal tyrosine-based sorting motif (YxxΦ) embedded in the TLR9 cytoplasmic tail allows phosphorylation and TLR9 targeting to late endosomes from which TNF expression is triggered in response to CpG ODN[41, 42]. We therefore took advantage of our capacity to perform RNA interference in CAL-1 cells and revisited these finding by investigating the impact of AP-3 silencing on human pDCs activation and BAD-LAMP transport, since it bears a 'YKHM' motif at the extremity (aa 276-280) of its cytoplasmic tail[26].

AP-3 was imaged together with TLR9 and BAD-LAMP in CpG-stimulated CAL-1 cells (Fig. 8a). TLR9 co-distribution with AP-3 was increased during the first hour of activation and remained nearly unchanged afterwards (Fig. 8a). BAD-LAMP distribution with the adaptor followed the same trend until its disappearance at 6 h (Fig. 8a). Efficient AP-3β (Supplementary Fig. 7A, B) silencing increased considerably BAD-LAMP (Fig. 8b) and TLR9 (Fig. 8c) co-distribution in VAMP3+ compartments, while TLR9 access to LAMP1-positive compartment, observed in control cells, was prevented (Fig. 8c). AP-3β silencing also largely impacted the formation of VAMP3+/LAMP2+ endo-lysosomes, compromising the induction and the clear segregation of this hybrid compartment, as judged by the overlapping distribution of all three endosomal markers used in the study (Supplementary Fig. 7C). AP-3β silencing also increased considerably TLR9 and BAD-LAMP co-localisation in both resting and CpG-stimulated cells suggesting that BAD-LAMP sorting is also affected by AP-3 deficiency (Supplementary Fig. 7D). As anticipated from TLR9 accumulation in VAMP3+ endosomes, CpG activation strongly augmented IFNα2 transcription in silenced cells (Fig. 8d), while TNF-α mRNA expression was reduced (Fig. 8d).

Given that BAD-LAMP bears a carboxyl-terminal YxxΦ motif, we also evaluated the importance of this AP-2/3 dependent sorting signal for TLR9 transport and signalling, by introducing a mutational change from tyrosine to alanine at position 276 in the cytoplasmic tail of BAD-LAMP (BAD-LAMP Y276A). We then compared the effect of ectopic expression of Flag-BAD-LAMP WT to Flag-BAD-LAMP Y276A on TLR9 localisation and signalling in CAL-1 cells. After electroporation of mRNAs coding for WT or mutant BAD-LAMP, protein levels (Flag staining) were maintained for at least 6 h after CpG treatment (Fig. 9a). BAD-LAMP Y276A accumulation in VAMP3 endosome was strongly enhanced compare to the WT molecule and its transport to LAMP1 endosomes was completely prevented (Fig. 9b). In agreement with this result, BAD-LAMP Y276A expression had no consequences on both TLR9 localisation (Fig. 9c) and cytokines expression (Fig. 9d), thus acting as a dead mutant.

Taken together, those results strongly indicate that the 'YKHM' motif of BAD-LAMP is necessary for its transport from VAMP3+ endosomes to late endosomes and the inhibition of IRF7 recruitment by TLR9 (Fig. 5d). Moreover, in human pDCs, AP-3 is necessary to maintain a normal endosome dynamic and compartmentalisation that is required by TLR9 and probably BAD-LAMP to reach LAMP1+ endosomes. Notably, increased

**Fig. 7** BAD-LAMP expression promotes TLR9-dependent TNF expression. CAL-1 cells were electroporated with BAD-LAMP mRNA (overexpressed BAD-LAMP) or control non-relevant mRNA (NT) during 6 h before stimulation during indicated times with CpG. **a, b** (*left*) Immunofluorescence confocal microscopy (ICM) on CAL-1 electroporated with BAD-LAMP mRNA or not and stimulated with CpG-A for indicated times. Staining for LAMP1, VAMP3 and BAD-LAMP (**a**) or TLR9 (**b**) is shown. Pictures are representative of at least three independent experiments. *White arrowheads* identify co-localisation area. *Scale bars* = 5 µm. (*right*) Quantification of the co-localisation between BAD-LAMP (**a**) or TLR9 (**b**) and VAMP3 (*red lines*) or LAMP1 (*blue lines*) between control (*full lines*) and transfected cells (*dashed lines*) upon CpG-A stimulation was performed by Pearson's coefficient measurement using ImageJ. Graphics represents mean of Pearson's coefficient of at least 25 cells ± s.d. for any time point. **c, d** IFNα2 (*left*) and TNF (*right*) mRNA (**c**) or protein (**d**) level were monitored by RT-QPCR or ELISA respectively. **e** (*top*) Voxel gating analysis after ICM on control (NT) or transfected cells with BAD-LAMP mRNA and treated during 3 h with CpG-A. Voxel gating was performed on TLR9 (*green*) and VAMP3 (*red*) channels to generate a 'Coloc channel' picture only showing VAMP3+ TLR9-containing endosomes. This new 'Coloc channel' was then merged with single IRF7 staining. *Arrowhead* indicates co-localisation area. *Scale bars* = 5 µm. (*bottom*) Quantification was performed by Pearson's coefficient measurement using ImageJ. Graphics represent Pearson's coefficient means of 25 different cells ± s.d. from two independent experiments. **f** (*left*) control CAL-1 cells or expressing FLAG-BAD-LAMP mRNA were treated with CpG-A for indicated times prior lysis and sodium dodecyl sulphate–polyacrylamide gel electrophoresis treatment. Expression of FLAG, AKT, IRF7, p65 NF-κB subunit and their phosphorylated forms were detected by immunoblot. β-actin is shown as loading control. (*right*) Quantification of IRF7 (*top*) and p65 (*bottom*) phosphorylation levels normalised to their total form by ImageJ quantification. *$P < 0.05$ by unpaired student's *t*-test

distribution of TLR9 in VAMP3[+] endosomes always correlates with stronger early type-I IFN expression and reduced TNF-α mRNA levels. A role for LAMP1[+] late endosomes or lysosomes as signalling hubs capable of promoting IFN expression in human pDCs in response to CpG seems however excluded by our results (Fig. 5).

**BAD-LAMP prevents type I IFN production in tumour pDCs.** Given the role of BAD-LAMP in controlling TLR9 signalling through endosomes sorting, we wondered whether immunomodulatory conditions known to influence pDCs function could interfere with BAD-LAMP expression or activity. Human pDCs infiltrating tumours can be functionally impaired and contribute

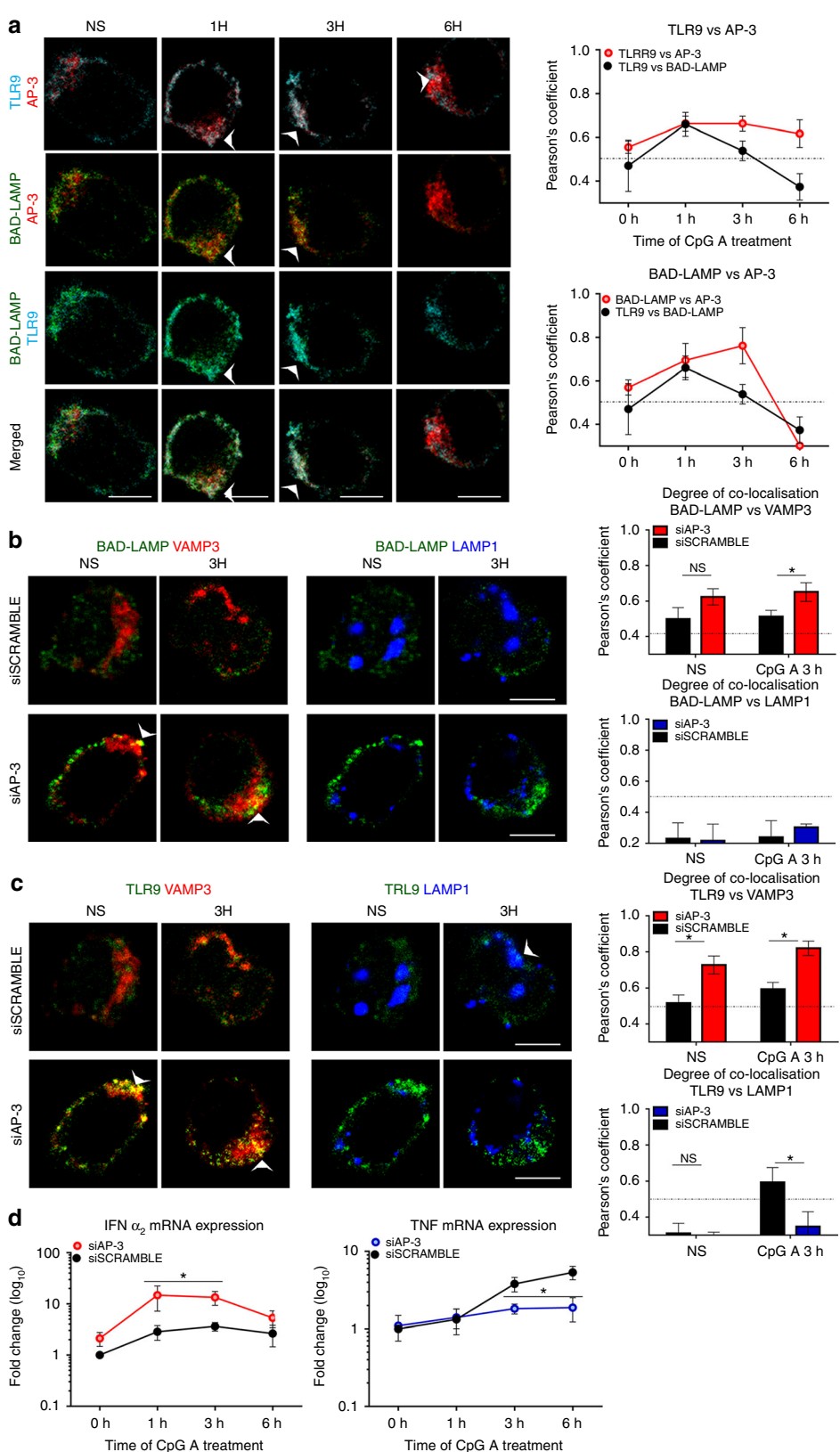

to the abnormal regulatory T cells amplification associated with poor prognosis[29, 43]. This negative impact of breast tumour-associated pDCs (TA-pDCs) on clinical outcomes has been linked to an IFN-α production defect driven by tumour-derived soluble factors, like transforming growth factor-β (TGF-β)[44]. Using BDCA2 and CD123 as selective markers, we quantified comparatively BAD-LAMP levels in pDCs isolated from the tumour and from the peripheral blood of the same patients (Fig. 10a). By flow cytometry analysis, BAD-LAMP levels were found strongly up-regulated in breast TA-pDCs compared to their blood counterparts. As BAD-LAMP levels correlated with the dysfunctional status of TA-pDCs, we examined the consequences of treating pDCs in vitro with tumours supernatants (snTUM) known to have or not immunosuppressive activity (Fig. 10b). Short-term treatment of blood pDCs with immunosuppressive supernatants containing TNF-α and high levels of TGF-β (snTUM+)[44], prevented BAD-LAMP downregulation in response to CpG activation, unlike treatment with non-immunosuppressive supernatants (snTUM-). When tumour supernatants were substituted by a combination of purified TGF-β and TNF-α, similar results were obtained with CpG-activated CAL-1 cells. Next, a combination of purified TGF-β and TNF-α was used to replace immunosupressive tumour supernatants with similar consequences on CAL-1 cells activation by CpG, thus confirming that TGF-β prevents the down-modulation of BAD-LAMP in activated pDCs (Supplementary Fig. 8A, B). Not surprisingly, type-I IFN mRNA expression was strongly inhibited in these cells (Figs 10c, d, Supplementary Fig. 8C), while TNF production remained however unaffected in the first hours of activation (Figs 10c, d and Supplementary Fig. 8C). TLR9 triggering by CpG was therefore not completely impaired by TGF-β or snTUM+ exposure, which impacted mostly type-I IFN expression. Given that BAD-LAMP reduces the capacity of TLR9 to trigger type-I IFN transcription, we hypothesised that a negative amplification loop initiated by TGF-β exposure and by reinforced BAD-LAMP expression could explain partly the dysfunctional status of tumour-associated pDCs. We therefore silenced BAD-LAMP in CAL-1 cells exposed to the immunosuppressive cytokines cocktail, and monitored their capacity to respond to CpG (Fig. 10e). BAD-LAMP silencing prevented the complete extinction of type-I IFN expression in response to CpG normally caused by TGF-β, and allowed TGF-β-treated cells to maintain IFN-α2 transcription levels equivalent to scramble control, while TNF expression remained unaffected (Fig. 10e). Taken together, these results again underline the importance of BAD-LAMP expression in controlling TLR9 signalling and suggest that its reinforced expression in tumour-associated pDCs contributes to their immunomodulatory phenotypes by decreasing the type-I IFN production capacity.

## Discussion

Plasmacytoid dendritic cells are important actors of antiviral immunity that produce large amounts of type-I IFN in response to nucleic acid detection by TLR7 and TLR9. Another fascinating aspect of pDCs biology is their ability to develop into tolerogenic DCs, driven by endogenous factors including cytokines and T-cell derived signals that may constitute an important self-tolerance mechanism during tumour development. The nature and the dynamic of the endosomes serving as assembly hubs for different TLRs signalling complexes is the driving force behind pDC's response to nucleic acids. Human pDCs have adopted a dynamic endosomes organisation, which favours intense type-I IFN production by allowing rapid signalling from early endosomal compartments. The inducible and transient existence of these chimeric organelles suggests that they could be considered as functional intermediates between sorting endosomes and LRO[45]. Such compartments present a mixed identity failing to completely segregate in specific endosomal subsets and are the results of active endosome reorganisation, as previously reported for MHC II-containing organelles in activated conventional DC[46] or in HeLa cell[47]. Hybrid compartments seem therefore to be a feature of cells controlling tightly their endosomes dynamic in response to immune-regulatory cues and could be key to modulate the balance and the strength of the signalling required to optimise cytokine production.

Our data suggest that AP-3 participates in this endosomal reorganisation and allows TLR9 and BAD-LAMP to access late endosomes from which TNF expression can be induced. Given the massive disturbance in endosome dynamics caused by AP-3 silencing, it would be extremely speculative to further assign a specific role for this adaptor in TLR9 sorting. However, we could not observe any early type-I IFN production impairment in AP-3-silenced CAL-1 cells, confirming that TLR9 signalling from VAMP3+ organelles is sufficient to drive early IFN-α expression in pDC. These observations are in agreement with the fact that not all CpG ODN types can trigger type-I IFN production, a capacity mostly reserved to multimeric CpG ODNs accumulating in early endosomes, such as CPG-A[39]. The bimodal mode of TLR9 triggering that allows NF-κB activation to be initiated from LAMP1+ late endosomes several hours post CpG-A exposure, suggests that these signalling cascades have evolved to present different thresholds of activation in various cell types and perhaps species. The equilibrium and the signalling strength emanating from these pathways is also strongly influenced by the immunological environment, such as pre-exposure to type-I IFN, TNF or TGF-β, which we have shown here all impact BAD-LAMP expression and consequently the polarity of TLR9 signalling.

The mutually exclusive co-localisation of UNC93B1 and TLR9 with BAD-LAMP in endosomes suggests a sequential mode of interaction during which, association with UNC93B1 allows TLR9 addressing to VAMP3+ endosomes, before BAD-LAMP-dependent sorting. In turn BAD-LAMP facilitates TLR9 transport to late endosomes and potentially increases its degradation, while inhibiting the signalling cascades leading to type-I IFN production. We have shown that BAD-LAMP and UNC93B1 are capable of co-localising together upon overexpression[28], whether

**Fig. 8** AP-3 is necessary for TLR9 sorting to LAMP1 lysosomes and TNF induction. **a** (*left*) Immunofluorescence confocal microscopy (ICM) on CAL-1 stimulated with CpG-A for indicated times. Staining for BAD-LAMP, TLR9, and AP-3 is shown. Pictures are representative of at least three independent experiments. *Arrowheads* indicate co-localisation area. *Scale bars* = 5 μM. (*right*) Quantification of the co-localisation between TLR9 and BAD-LAMP (*black line* as reference), AP-3 and BAD-LAMP (*top graph red line*), and AP-3 and TLR9 (*bottom graph red line*) at different time was performed by Pearson's coefficient measurement using ImageJ. **b**, **c** (*left*) Immunofluorescence confocal microscopy on CAL-1 electroporated with AP-3 siRNA or scramble siRNA and stimulated with CpG-A for indicated times. Stainings for LAMP1, VAMP3 and BAD-LAMP (**b**) or TLR9 (**c**) are shown. These pictures are representative of at least three independent experiments. *White arrows* identified co-localisation area. *Scale bars* = 5 Mm. *right* Quantification of the co-localisation between BAD-LAMP (**b**) or TLR9 (**c**) with VAMP3 (*top*) or LAMP1 (*bottom*) at steady state (NS) or 3 h upon CpG-A stimulation was performed by Pearson's coefficient using ImageJ. Graphics represent Pearson's coefficient means of 50 different cells ± s.d. from at least three independent experiments. **d** IFNα2 (*left*) and TNF (*right*) mRNA level were monitored by RT-QPCR. Raw data have been normalised to housekeeping gene (GAPDH) and graphics represent fold change ± s.d. compare to non-stimulated cells from three independent experiments minimum. *$P < 0.05$ by unpaired student's t-test

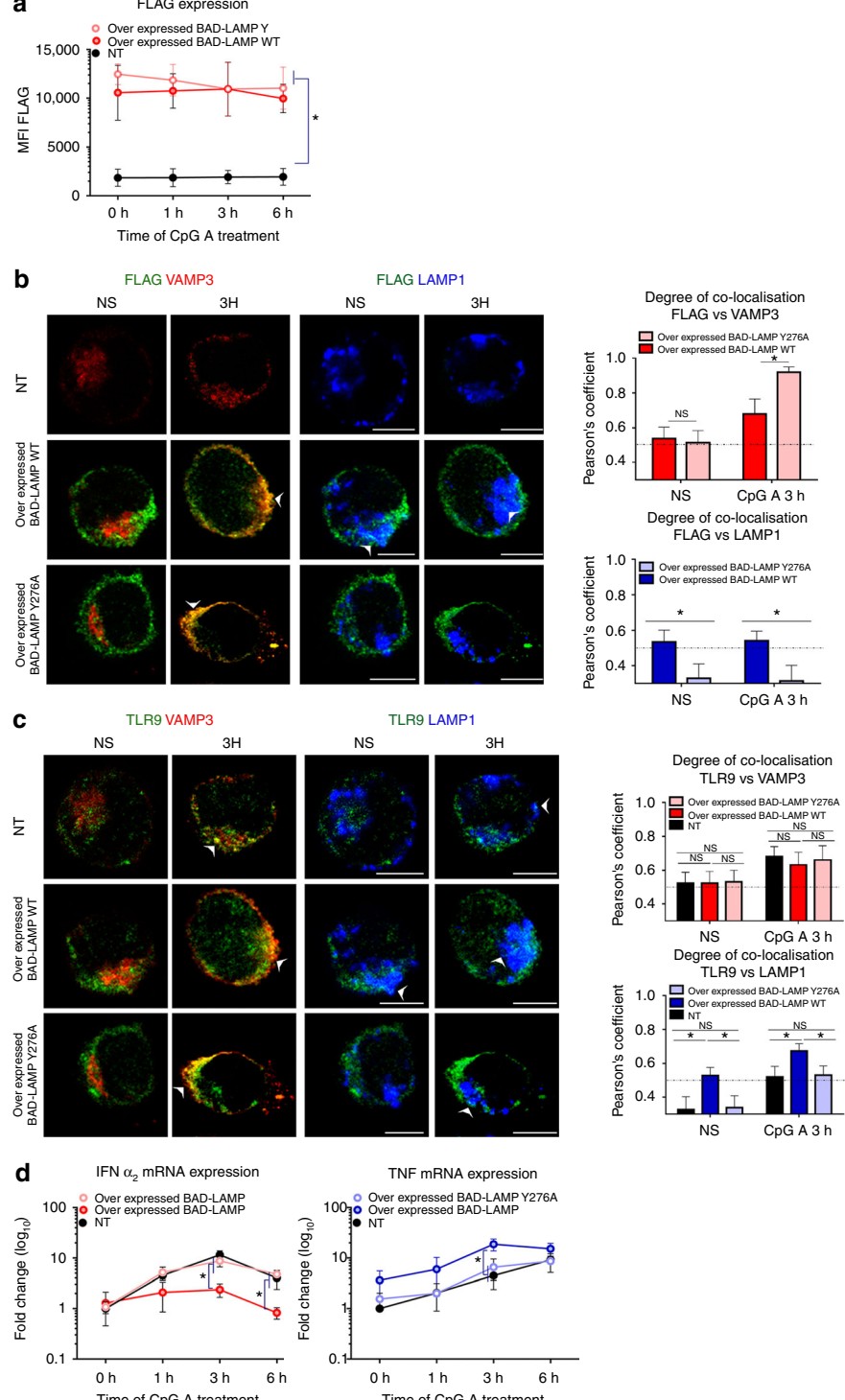

**Fig. 9** BAD-LAMP YxxΦ motif is required for TLR9 transport to LAMP1⁺ late endosomes. CAL-1 cells were electroporated with FLAG BAD-LAMP WT or FLAG BAD-LAMP Y276A mRNAs for 6 h prior stimulation for indicated times with CpG-A. Mock electroporated cells (NT) are used as control. **a** BAD-LAMP ectopic protein expression were monitored by intracellular flow cytometry with FLAG tag antibody. Graphic represents means of MFI ± s.d. from at least two independent experiments. **b**, **c** (*left*) Immunofluorescence confocal microscopy on CAL-1 cells. Staining for LAMP1, VAMP3, FLAG (**b**) or TLR9 (**c**) are shown. Pictures are representative of at least two independent experiments. *White arrows* identify co-localisation area. *Scale bars* = 5 μM. (*right*) Quantification of the co-localisation between FLAG (**b**) or TLR9 (**c**) with VAMP3 (*top*) or LAMP1 (*bottom*) at steady state (NS) and 3 h after CpG-A stimulation was performed by Pearson's coefficient measurement using ImageJ. Graphics represents mean of Pearson's coefficient of at least 25 cells ± s.d. for all time points. **d** IFNα₂ (*left*) and TNF (*right*) mRNA level were monitored by RT-QPCR. Raw data have been normalised to housekeeping gene (GAPDH) and graphics represent fold change ± s.d. compare to non-stimulated cells from two independent experiments. *$P < 0.05$ by unpaired student's *t*-test

this co-localisation is exclusive and interferes with binding to endocytic TLRs remains to be clarified. Until now, most of the biochemistry and cell biology performed on TLR9 and its inter-actors have been carried-out using overexpressed tagged-proteins (in particular TLR9-GFP) in non-relevant cell lines. Given that quality antibodies allowing a good human TLR9 biochemical

analysis are not available, we had to turn to advanced microscopy techniques to evaluate the interactions of BAD-LAMP, UNC93B1 and TLR9 in primary human pDCs. iPLA was of great efficacy to demonstrate the close vicinity and likely interactions of these different endogenous molecules during time, however until now our attempts to demonstrate unequivocally a physical binding

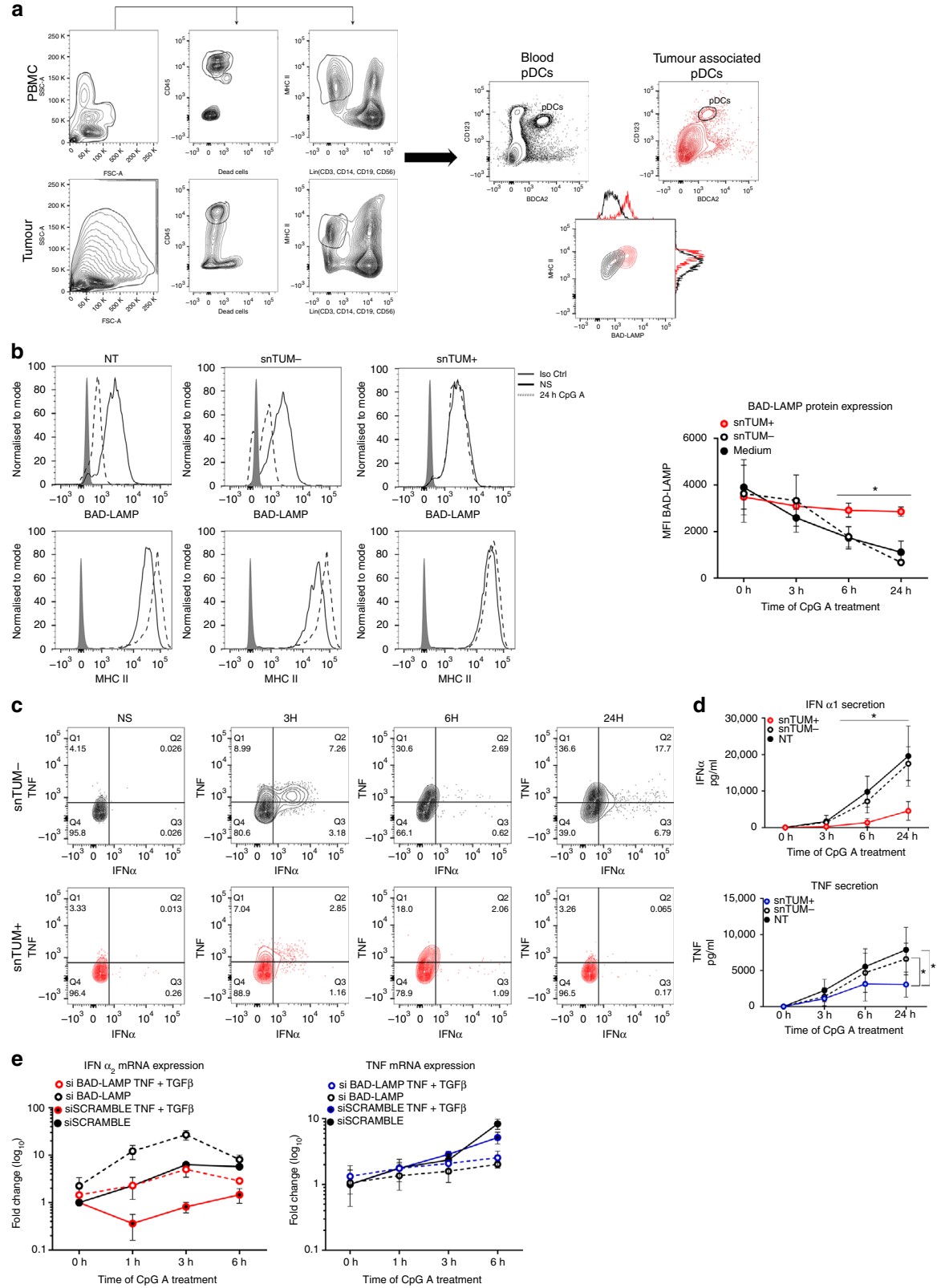

using traditional co-immunoprecipitation methods remained unsuccessful.

Using UNC93B1 silencing, we however showed that BAD-LAMP depends on this chaperone and/or pDC activation to reach massively VAMP3+ endosomes with TLR9 (Fig. 6a), suggesting that UNC93B1 and BAD-LAMP interaction in the ER or ERGIC could be part of the regulatory process controlling TLR9 transport in pDC. Interestingly, TLR9 accumulation at steady state in VAMP3+ compartments upon BAD-LAMP-silencing, further suggests that a constant but limited flow of BAD-LAMP and TLR9 molecules can reach endosomes independently of pDC activation. Upon DNA detection, this TLR9 pool could be responsible for the first signalling events initiating the massive UNC93B1-dependent endosome recruitment of ER-resident TLR9 required for potent type-I IFN production. BAD-LAMP expression seems however to limit the activation of this pool, presumably by promoting TLR9 transport to late endosomes, whose degradative environment could increase the threshold of TLR9 activation by endogenous ligands, compared to the situation in VAMP3+ compartments.

To preserve the organism from the inherent toxicity of recurrent and excessive release of type I IFN, it is crucial that pDCs avoid inappropriate activation by circulating self-DNA, but still respond to microbial cues[48]. This issue is particularly relevant to human pDCs, since their extreme specialisation as professional IFN-producers is dwarfing their mouse counterparts. It has been proposed that responses to self-DNA are limited by the requirement for endocytosis and residency time with TLR9/MYD88 complexes in early endosomes[19, 49]. The capacity of BAD-LAMP to decrease TLR9 capacity to drive IFN responses suggests that human pDCs require additional safeguards mechanisms to limit abnormal type-I IFN production and regulate their capacity to produce pro-inflammatory cytokines. BAD-LAMP expression seems to maintain a molecular safety lock on TLR9 signalling by reducing its capacity to produce spontaneously type-I IFN. During the early phase of activation, BAD-LAMP allows TLR9 to reach late endosomes from which NF-κB activation can occur, if appropriate quantities of nucleic acid ligands are present in these compartments. If pDC exposure to type-I IFN prevails, BAD-LAMP down-regulation facilitates TLR9 retention and signalling from VAMP3+/LAMP2+ compartments allowing a positive-amplification loop increasing IFN production. BAD-LAMP expression levels are therefore correlated with the reduced capacity of pDCs to produce type-I IFN and increased potential to develop into tolerogenic DCs. This particular feature of BAD-LAMP seems to be exploited in tumour infiltrating pDCs, since higher expression of BAD-LAMP in pDCs exposed to tumour microenvironments contribute to their functional impairment and incapacity to produce type-I IFN, both associated with poor prognosis. BAD-LAMP therapeutic modulation could therefore represent a novel strategy for harnessing pDCs responses in different clinically relevant contexts such as autoimmunity or tumour development.

## Methods

**Patients, human tissue samples and blood.** Fresh tumours from patients diagnosed with primary breast carcinoma were obtained prior any treatment from the Centre Léon Bérard (CLB) tissue bank after patient informed consent. The study was reviewed and approved by the Institutional Review Board of CLB, Lyon. Discarded human tonsil material was obtained anonymously according to the institutional regulations in compliance with French law. Written informed consent was obtained from all study participants in accordance with the Declaration of Helsinki. Healthy human blood and Buffy coats were obtained by leukapheresis (Etablissement Français du Sang (EFS) Marseille, France). The CAL-1 cells were the kind gift of Dr. Takahiro Maeda from Nagasaki University Graduate school of Biomedical Science, Japan.

**Isolation and stimulation of pDCs.** Mononuclear cell enriched from blood of healthy donors was submitted to Ficoll-paque plus and percoll gradient (all from PBL Biomedical Laboratories). The pDCs were MACS sorted using a negative selection kit as per manufacturer's instructions (plasmacytoïd dendritic cells II Miltenyi biotec). The pDCs isolated by this procedure were 93–98% pure and their viability was > 95%. Healthy pDCs were cultured in the presence of IL-3 (1ng/ml) and stimulated with CpG-A ODN 2216 (Tebu bio) for indicated time. Cell culture with breast tumour supernatant (TumSN) was performed as described in supplementary methods[44]. Cal-1 cells (the kind gift of Dr. T. Maeda Nagasaki University, Japan) were grown in RPMI (GIBCO), 10% FCS (Sigma Aldrich), 2mM L-glutamine, 1x non-essential amino acids, 10 mM HEPES, 1 mM sodium pyruvate (all the previous from GIBCO). For performing experiments, cells were plated 16 h before in a 12 wells plate at $1 \times 10^6$ cells/ml, 1 ml × well, in a complete medium with 1% FCS. All cell lines were mycoplasma free and kept at 37℃ and 5% $CO_2$.

**Detection of cytokines.** ELISA kit for human type I IFN dosage in CAL-1 supernatants was from PBL Biomedical Laboratories. ELISA kit for human IFNα$_1$ dosage in freshly isolated pDCs supernatants and for human TNF were from eBiosciences. All were used according to manufacturer's instructions and described in Supplementary Table 2.

**Proximity ligation assays.** CAL-1 and freshly isolated pDCs were activated for the indicated time and treated as for conventional immunofluorescence microscopy. Secondary PLA probes were from Sigma and ligation and Red Amplification solutions were used according to manufacturer's instructions. Washed cells were mounted on microscopy slide using Duolink II Mounting Medium with DAPI. Image stacks were captured with an inverted confocal microscope Zeiss LSM580 or a Leica TCS SPX5, using 63× objective and accompanying imaging softwares described in Supplementary Table 2. The images were quantified using the analyse particles plugin of ImageJ.

**Antibodies and immunodetection.** 25–50 µg of NP-40 soluble material was separated by 10% sodium dodecyl sulphate–polyacrylamide gel electrophoresis prior immunoblotting and chemiluminescence detection (Pierce), original uncropped immunoblots are shown in Supplementary Figs 9–13. Antibodies used in this study were: Rat mAb 34.2 anti-BAD-LAMP (LAMP5)[26], rabbit polyclonal

**Fig. 10** BAD-LAMP expression in tumour-associated pDCs correlates with low IFN-α production. **a** (*left*) Flow cytometry pDC gating strategy on both PBMC (*top*) and tumour (*bottom*) is shown. pDCs (BDCA2+; CD123+) were segregated from conventional dendritic cells (Lin−, MHC II high) after identification of live hematopoietic cells (CD45+ Live Dead−). (*right*) CD123+/BDCA4+ pDCs from blood (*black*) or from primary breast tumours (*red*) were further analysed for BAD-LAMP and MHC II expression. Data are representative of three patients. **b** Freshly isolated pDCs from healthy donors were treated for 16 h with tumour supernatant either devoid of (snTUM−) or enriched in (snTUM+) TNF and TGF-β. (*left*) BAD-LAMP and MHC II histogram plots from FACS staining at steady state (*black line*) or 24 h (*dashed line*) after CpG-A treatment. *Full grey* histogram represents isotype control. Data are representative for five independent experiments performed with eight different tumour supernatants. (*right*) Levels of BAD-LAMP shown as MFI from FACS intracellular staining in pDCs pre-treated with snTUM- (*dashed line*), snTUM + (*red line*) or with medium (*black lines*). MFI ± s.d. from five independent experiments. **c** Freshly isolated pDCs from heathy donors were treated for 16 h with tumour supernatant either devoid of (snTUM-; *black*) or enriched in (snTUM +; *red*) in TNF and TGF-β. 2D FACS analysis of intracellular staining for IFN-α and TNF cytokines at different times after CpG-A stimulation. Data are representative for three independent experiments. **d** IFN-α (*top*) and TNF (*bottom*) production was monitored by ELISA at different times of CpG-A stimulation. Graphics represent cytokines concentration ± s.d. from five independent experiments minimum. Eight different sets of tumour supernatants have been tested. **e** IFNα$_2$ (*left*) and TNF (*right*) mRNA expression in CAL-1 electroporated with BAD-LAMP siRNA (*dashed lines*) or scramble siRNA (*full lines*) and stimulated for indicated times with CpG-A. CAL-1 were treated either with TNF + TGF-β (*coloured*) or with medium (*black*). Raw data have been normalised to housekeeping gene (GAPDH) and graphics represent fold ± s.d. compared to non-stimulated cells from two experiment. *$P < 0.05$ by unpaired student's t-test

anti-, phospho-p65, phospho-S6, S6, phospho-IRF7, phospho-TBK1 were from Cell Signalling Technology, anti-IRF7 was from Santa Cruz, anti-p65, anti-TLR9 (H-100), anti-UNC93B1 (E-12) were all from Santa Cruz biotechnology. Anti-VAMP3 was the kind gift from Thierry Galli (Institut J. Monod, Paris). Anti-flag was purchased from Sigma. Goat polyclonal anti-VAMP3 (N12) was from Santa Cruz biotechnology and anti-MYD88 from R&D System. Anti-Mouse monoclonal anti-actin (AC-15), anti-Syntaxin 6 and anti-AP-3 were all from BD transduction laboratories. Brilliant violet 421 and anti-LAMP1 (H4A3) were from Biolegend. Alexa fluor 647 anti-LAMP2 (H4A3) were from Molecular probes. Secondary antibodies were from Jackson Immunoresearch, Molecular Probes and Cell Signalling Technology. For immunofluorescence, cells seeded on alcian blue-treated coverslips were fixed with 3.5% PFA and permeabilized with 0.05% saponin. Immunofluorescence and confocal microscopy was performed with a Zeiss LSM580 or a Leica TCS SPX5 using a 63 x objective and accompanying imaging softwares described in Supplementary Table 2. Fluorochrome- or biotin-coupled antibodies for FACS were CD123 (6H6), anti TNF-alpha (MAb11), LAMP5 (34.2) were all from eBioscience. Anti-CD303 (201A), anti-CD304 (12C2) were from Biolegend and anti-IFNα (LT27:295) from miltenyi biotech. All antibodies are described in Supplementary Table 1. Data was acquired on a FACS LSR II or Canto II (BD Biosciences) and analysed using FlowJo (Treestar). Anti-Type 1 IFN-receptor antibody 5 µg/ml (mouse anti-human IFNα/βR chain 2) was used for inhibition in vitro.

**Molecular biology.** All plasmids were generated using standard cloning, PCR and fusion PCR techniques and based on pcDNA3.1 backbone (Invitrogen)[28]. For *in vitro* transcription, a BAD-LAMP cDNA WT and Y276A were introduced in the PGEM4Z vector (Promega) and transcribed in vitro by using mMessage T7 machine (Thermo Fisher) and poly adenylated with poly A tail kit (Thermo Fisher). All procedure has been performed according manufacturer's instructions and described in Supplementary Table 2. 4 × 106 CAL-1 cell were transfected with 5 ug of mRNA using the Amaxa Cell line Kit V nucleofection kit according the manufacturer's instructions and described in Supplementary Table 2. CAL-1 were activated 6 h post transfection for the indicated time and analysed. FlexiTube siRNA targeting BAD-LAMP, UNC93B1, AP-3b1 and FlexiTube siRNA scramble as control (QIAGEN) were used for RNA silencing experiments. 4 × 10^6 CAL-1 cell were transfected with 1 ug of four siRNA mix using the Amaxa Cell line Kit V nucleofection kit. CAL-1 were activated 24 h post transfection for the indicated time and analysed. For quantitative PCR, total RNAs were extracted and purified using the RNeasy Mini Kit (Qiagen). 100 ng to 1 µg of total RNA were subjected to reverse transcription using SuperScript II. Each gene transcripts were quantified by SYBR Green method with 7500Fast (Applied Biosystems). The relative amount of each transcript was determined by normalising to internal housekeeping gene expression. Oligonucleotides primers are described in Supplementary Table 3.

**Statistical and quantitative analysis.** Pearson coefficient was calculated with ImageJ using the JACOP plugging on the whole cell image. For each condition between 25 and 50 cells were analysed. Data representing multiple experiments is displayed as mean ± s.d. UnPaired student's *t*-test was used (Holm–Sidak method) where indicated. Voxel gating were performed with the 'Coloc' tool from the IMARIS software between 25 and 50 cells were analysed per conditions.

**Data availability.** The data that support the findings of this study are available from the corresponding author upon request.

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

## Acknowledgements

A.C. received a MNERT and A.R.C. PhD fellowship. P.n'G. is supported by the Ligue Nationale Contre le Cancer (LNCC). The laboratory is supported by grants from L'Association de la Recherche contre le Cancer (ARC) to E.G., from l'Agence Nationale de la Recherche (ANR), «ANR-12-BSV2-0025-01», «ANR-FCT 12-ISV3-0002-01», «INFORM Labex ANR-11-LABX-0054», «DCBIOL Labex ANR-11-LABEX-0043» and ANR-10-IDEX-0001-02 PSL* and A*MIDEX project ANR-11-IDEX-0001-02 that are funded by the Excellence Initiative of Aix-Marseille University-A*MIDEX, a French 'Investissements d'Avenir' program. N.B.V and C.C. were financially supported by the Grant INCa PAIR SEIN 2014-093, the SIRIC project (LYRIC, grant no. INCa_4664). This work was performed within the framework of the LABEX DEVweCAN (ANR-10-LABX-0061) of the University of Lyon, within the program 'Investissements d'Avenir' (ANR-11-IDEX-0007) organised by the French National Research Agency (ANR). We acknowledge support from no. ANR-10-INBS-04-01 France Bio Imaging for advanced microscopy. Part of the research was also supported by FCT through the Institute for Biomedicine – iBiMED contract UID/BIM/04501/2013 and PTDC/IMI-IMU/3615/2014 and the Fundação Ilídio Pinho.

## Author contributions

A.C., N.B.-V., C.C., P.P. and E.G. designed research and analysed data. A.C. V.C., P.N'G., R.J.A., J.M. performed research. A.C., N. B.-V., P.P. and E.G. wrote the paper.

## Additional information

**Competing interests:** The authors declare no competing financial interests.

