## [Peer review file · Nature Communications]

Reviewers' comments:

Reviewer #1 (Remarks to the Author):

This manuscript entitled "BAD-LAMP controls TLR9 trafficking and signaling in human plasmacytoid dendritic cells" shows that BAD-LAMP, whose expression is specifically regulated by Type I Interferon (IFN) and TGF- β in tumor microenvironment, co-localized with TLR9 and MyD88. BAD-LAMP controls TLR9 trafficking from Vamp3-positive compartment to Lamp1-positive compartment to regulate TLR9 signaling. In human pDCs, there is a bimodal activation of TLR9 signaling pathways inducing type I IFN in an IRF7-dependent manner, followed by proinflammatory cytokines in an NF- κ B-dependent manner. The imaging results are well organized and nicely shown. However, the following questions remain. More detailed analyses are needed.

1. Authors analyze TLR9 localization by anti-human TLR9 polyclonal antibody. It is important to show the specificity of this antibody by staining TLR9-knockdown pDC cell line.
2. In figure 3-6, authors claimed the formation of the hybrid compartment containing Vamp3, Lamp2 and TLR9 after CpG-A stimulation. However, data is not fully analyzed to prove the hybrid compartment. Authors should analyze the colocalization of Vamp3, Lamp2, and TLR9 by voxel gating and coloc channel with statistical analyses. Overexpression of BAD-LAMP decreased IFN- α expression but increased TNF- α expression (Fig. 5).
3. If early endosome serves as IRF-signaling endosome as suggested in Fig. 4 and 5, TLR9 and Vamp3 colocalization is expected to decrease by BAD-LAMP overexpression. However, TLR9 and Vamp3 colocalization was unaltered despite BAD-LAMP overexpression (Fig. 5). The authors need to explain this inconsistency.
4. In Fig. 4 and 5, time course of phosphorylation of IRF7 and p65 in TLR9 signaling should be analyzed. The reviewer expects that IRF7 phosphorylation becomes longer/stronger in Fig. 4 and shorter/weaker in Fig. 5.
5. BAD-LAMP seems to regulate TLR9 trafficking from Vamp3 positive compartment to LAMP1 positive compartment. Signaling for TNF- α expression seems to start at LAMP1 positive compartment. CpG-A stimulation downregulated BAD-LAMP expression (Fig. 1). Why TLR9 trafficked to LAMP1 positive compartment despite the low level of BAD-LAMP? Low BAD-LAMP is expected to hold TLR9 in Vamp3-positive compartment, IRF endosome. Authors should discuss this issue.
6. If possible, authors should show direct evidence that the hybrid compartment is the IRF-signaling endosome by colocalization of TLR9 and IRF7 in the hybrid compartment.

Minor comment

1. Figure-1D, phospho-p65 signal disappeared at 3h after CpG stimulation. This is probably due to a problem during western blotting. The results should be replaced. In addition, authors should analyze phospho-p65 at earlier time point within 60 min to exclude a possibility that NF- κ B activation occurred within 60 min.
2. Figure-S1, type I IFN treatment decreased BAD-LAMP expression more significantly compared with CpG-A stimulation by FACS analysis. However the MFIs show that CpG-A was more effective than type I IFN in decreasing BAD-LAMP expression. The authors need to explain this inconsistency or replace these results.
3. In confocal analysis, blue and green analysis is difficult to see. Authors should change to green and red analysis or other colors to make the results easier to understand.
4. Figure-2B, Pearson Coefficient analysis is needed.
5. Figure-2C, BAD-LAMP and UNC93B1 seems to be colocalized at non-stimulation and 1h stimulation especially in CAL-1 cell. But the statistical analyses showed the decrease in the colocalization at 1 h after stimulation. The authors need to explain this inconsistency or replace the results. Scale bars are also needed.
6. In figure-S4B, the colocalization of TLR9 and Vamp3 looks same before and after CpG-A stimulation in siBAD-LAMP cells in images. However, the statistical analyses indicate the increase of the colocalization by CpG-A stimulation. The authors need to explain this inconsistency or

replace the results.

7. In all the statistical analyses, Pearson's coefficient analysis should show from 0 to 1 at Y axis.

8. The length of scale bars and P values are often missing. They need to be described.

Reviewer #2 (Remarks to the Author):

This manuscript reports on the role of the endo-lysosomal membrane protein BAD-LAMP and its impact on TLR9 trafficking, as well as downstream signaling events associated with it (cytokine production). In the absence of compelling biochemical data such as co-immunoprecipitation -the Pierre lab has state-of-the-art technology for these types of applications- claims of physical association cannot be sustained, as they are based solely on colocalization by immunofluorescence, a technique that lacks the resolution to claim direct interactions. The work is quite descriptive in nature, an obvious consequence of the technical approach chosen, and I doubt whether this work will influence thinking in the field, unless the fluorescence data were backed up by the molecular approach alluded to above. The proximity ligation assay samples too small a number of molecules (too few events) to allow an unambiguous interpretation of the proposed interactions.

The lab is expert in immunofluorescence, and I have no reason to question any of the results presented as micrographs. Electron micrographical support (immunoEM) would have been nice to provide a higher resolution overview but is clearly beyond the scope of this work at present. The quality of the files force the reader to accept many of the interpretation of the micrographs on faith. To the authors' credit, they apply methods to quantify colocalization of fluorescence signals by the 2D plots, a method with which the authors have a lot of experience, but these, too reproduced poorly. I found the discussion to lack clarity, with no clear "take home" message.

Minor points: The text still contains many gallicisms that should be weeded out, regardless of where this work would be published (inappropriate use of plural nouns as modifiers of other nouns, no agreement between plural subject and singular verb, words like 'spacio-temporal' "specie-specific" etc., "we could not observe" .

The term intracellular FACS is a misnomer, as no sorting of any kind is performed on cells

Reviewer #3 (Remarks to the Author):

To the Authors,

This is an impressive and technically challenging study with in-depth analysis at what is controlling TLR9 signaling in pDCs. The authors have used both primary cells or the cell line CAL-1 which although does not fully replicate the biology of pDC - in particular with low IFN response - allowed the authors to conduct more detailed mechanistic studies. I have the following concerns:

1- The authors observed that BAD-LAMP expression is reduced in pDCs following culture with CpG-A. This can be due to the presence of IFN induced by CpG-A, to TLR9 signaling or both. The authors show that IFN is enough to decrease BAD-LAMP but they should also show whether a CpG-B, which does not induce much IFN, can lead by itself to the same decrease and whether it then impacts the TLR9/MyD88/BAD complex formation.

2- The authors are repeatedly raising the idea of a sequential involvement of the IRF and NF-kB pathways but are only using CpG-A in their experiments which are known to be poor activator the NF-kB pathway. Using a CpG-C would have seemed to be a better choice and the authors should explain the rationale behind their choice of reagents.

3- The data presented in figure 5 are key to the conclusions presented by the authors as the data show that the overexpression of BAD-LAMP in CAL-1 cells impacts TLR9 distribution and IFN/TNF production. As the level of BAD-LAMP is drastically increased (Sup fig 5) and the level constant for 6h, how do the authors explain the relatively modest impact on IFN and TNF. This should be discussed.

4- It seems that the anti-TLR9 antibody used in the manuscript do not bind the TLR9 in all the organelles inside the pDCs or CAL-1 cells. One risk is that the altered staining could be due to non-specific binding. The authors should show a negative control using TLR9-negative cells.

5- In many of the figures, the authors are plotting IFN or TNF expression as fold changes. This can be misleading in particular as these 2 cytokines are not expressed at 0h. The authors should show the actual relative expression levels to the housekeeping gene.

6- As a minor comment, the last paragraph of the introduction is not a place to repeat the abstract and should not be used to just summarize the main findings.

Reviewer 1:

1. Authors analyze TLR9 localization by anti-human TLR9 polyclonal antibody. It is important to show the specificity of this antibody by staining TLR9-knockdown pDC cell line.

We provide for this reviewer's eyes a confocal image of TLR9 staining in HEK293 cells expressing or not TLR9, to demonstrate the specificity of the anti-TLR9 antibodies used in this study and the absence of background in cells non-expressing TLR9. We did not feel however that it was necessary to integrate this data set in the manuscript.

Control staining for anti-TLR9 (H100)
performed on HEK cells transfected (HEK TLR9) or not (HEKnull)

2. In figure 3-6, authors claimed the formation of the hybrid compartment containing Vamp3, Lamp2 and TLR9 after CpG-A stimulation. However, data is not fully analyzed to prove the hybrid compartment. Authors should analyze the colocalization of Vamp3, Lamp2, and TLR9 by voxel gating and coloc channel with statistical analyses.

We have now included an entirely new supplementary figure (new S3), focussing on this hybrid compartment. We show using voxel gating, how the dynamic of Vamp3, Lamp2, and TLR9 colocalization is evolving during the formation of this hybrid compartment, and that it is also the site of the preferential recruitment of IRF7 by TLR9. We have also reconstructed in 3D these compartments (using the IMARIS-based deconvolution software), and could clearly observe first a partitioning of LAMP1 and LAMP2 in distinct membrane domains of the same organelles, away from VAMP3+ compartments, demonstrating that a specific distribution of LAMP1 and LAMP2 pre-exists in the late endosomes of steady state pDC. pDC activation by CpG ODN, leads rapidly to an increase of this partitioning and the transient but intense formation of VAMP3/LAMP2+ hybrid organelles, prior returning at the steady state organization after 6h.

3. *Overexpression of BAD-LAMP decreased IFN- α expression but increased TNF- α expression (Fig. 5). If early endosome serves as IRF-signaling endosome as suggested in Fig. 4 and 5, TLR9 and Vamp3 colocalization is expected to decrease by BAD-LAMP overexpression. However, TLR9 and Vamp3 colocalization was unaltered despite BAD-LAMP overexpression (Fig. 5). The authors need to explain this inconsistency.*

One has to consider that even if TLR9 transport to Lamp1+ late endosomes is considerably increased by BAD-LAMP over-expression, a maximum transport capacity of these molecules to late endosomes is likely to be reached. Thus, an increased TLR9 accumulation in late endosomes does not reflect necessarily a complete abolition of TLR9 residency in VAMP3-positive endosomes, given the likely existence of a transport bottleneck from sorting endosomes to late endosomes. Importantly, in addition of affecting TLR9 transport, we showed that BAD-LAMP could interfere directly with IRF7 recruitment, thus explaining the strong inhibitory effect of BAD-LAMP expression on IFN production, despite the remaining presence of TLR9 molecules in the VAMP3-signaling endosomes. We have now added in Fig. 5 a novel panel (5E), supporting this view, and showing that BAD-LAMP expression interferes with IRF7 recruitment by TLR9 in VAMP3 endosomes. These data are complemented by iPLA microscopy approach showing in Fig. S5, that BAD-LAMP silencing enhances the recruitment of IRF7 by TLR9 on VAMP3 endosomes, while conversely BAD-LAMP overexpression (Fig. S6) limits considerably the interaction of TLR9 with IRF7.

4. *In Fig. 4 and 5, time course of phosphorylation of IRF7 and p65 in TLR9 signaling should be analyzed. The reviewer expects that IRF7 phosphorylation becomes longer/stronger in Fig. 4 and shorter/weaker in Fig. 5.*

We have now performed immunoblots to analyse the signalling kinetics in different experimental situations. upon BAD-LAMP silencing, the phosphorylation status of AKT, IRF7 and p65 in CAL-1 cells is now shown in a new panel Figure 4E. As expected by Reviewer 1, IRF7 activation is reinforced in absence of BAD-LAMP, while P-p65 is strongly decreased, in particular at steady state, confirming the microscopy data (Fig. S5). Interestingly, AKT activation seems also to be reinforced in the silenced cells, suggesting that AKT activation by TLR9 is also triggered in VAMP3-positive early endosomes, in agreement with the observation by Guiducci et al. (JEM 2008), that AKT activation occurs within 20 min of pDC stimulation by CpG or Flu. As for BAD-LAMP over-expression (new Fig 5F), IRF7 and AKT phosphorylation were nearly abolished, further confirming the inhibitory role of BAD-LAMP expression on TLR9-signaling from Vamp3-positive endosomes and on type-I IFN production. p65 phosphorylation levels were not dramatically increased compared to control, however its activation appeared to be more rapid upon BAD-LAMP expression, confirming the consequences of enhanced TLR9 accumulation in late endosomes.

5. *BAD-LAMP seems to regulate TLR9 trafficking from Vamp3 positive compartment to LAMP1 positive compartment. Signaling for TNF- α expression seems to start at LAMP1 positive compartment. CpG-A stimulation downregulated BAD-LAMP expression (Fig. 1). Why TLR9 trafficked to LAMP1 positive compartment despite the low level of BAD-LAMP? Low BAD-LAMP is expected to hold TLR9 in Vamp3-positive compartment, IRF endosome. Authors should discuss this issue.*

Using Bafilomycin treatment, we could demonstrate that BAD-LAMP is targeted to late endosomes and degraded rapidly upon pDC activation, as shown in a novel immunoblot presented in Fig S2. BAD-LAMP disappearance was inhibited by bafilomycin treatment, which also promoted BAD-LAMP accumulation in LAMP1⁺ late endosomes. Thus, BAD-LAMP is able to traffic from sorting to late endosomes and potentially bring TLR9 along, while preventing IRF7 recruitment in early endosomes. Importantly, BAD-LAMP disappearance is not immediate and could still favour the transport of TLR9 to late endosomes, once engaged in this pathway, TLR9 could traffic independently or receive assistance from other molecules like the recently characterized SCARB2/LIMP-2, that participates to TLR9 traffic to late endosomes in mouse pDCs (Guo et al. J. Immunol, 2015).

6. *If possible, authors should show direct evidence that the hybrid compartment is the IRF-signalling endosome by co-localization of TLR9 and IRF7 in the hybrid compartment.*

As mentioned above (point 2), we have performed this analysis both using voxel gating and iPLA (Fig S5) confirming the importance of the hybrid compartment for IRF7 signalling.

Minor

comment

1. *Figure-1D, phospho-p65 signal disappeared at 3h after CpG stimulation. This is probably due to a problem during western blotting. The results should be replaced. In addition, authors should analyze phospho-p65 at earlier time point within 60 min to exclude a possibility that NF- κ B activation occurred within 60 min.*

This has been done and the results shown in Figure 1D confirm our original observation.

2. *Figure-S1, type I IFN treatment decreased BAD-LAMP expression more significantly compared with CpG-A stimulation by FACS analysis. However, the MFIs show that CpG-A was more effective than type I IFN in decreasing BAD-LAMP expression. The authors need to explain this inconsistency or replace these results.*

We do not think there is a significant difference between the two treatments, the flow cytometry profile has now been replaced with a new analysis consistent with the MFI of the pooled results (n=3).

3. *In confocal analysis, blue and green analysis is difficult to see. Authors should change to green and red analysis or other colours to make the results easier to understand.*

We have taken great care of the presentation and choice of the different colors. We are aware the multiple staining is sometimes difficult to visualize, and the chosen combination of colors is, in our opinion, the best possible. The quantifications provided for all microscopy panels should facilitate results readability and interpretation.

4. *Figure-2B, Pearson Coefficient analysis is needed.*

We did not understand this comment, as the coefficient analysis was displayed in the same figure.

5. *Figure-2C, BAD-LAMP and UNC93B1 seems to be colocalized at non-stimulation and 1h stimulation especially in CAL-1 cell. But the statistical analyses showed the decrease in the colocalization at 1 h after stimulation. The authors need to explain this inconsistency or replace the results. Scale bars are also needed.*

We do observe iPLA staining for UNC93-B1 and BAD-LAMP in CAL-1 cells after 1h of stimulation, however the number of dots per cell at this time point is strongly reduced compared to the non-treated samples. In our eyes the difference is striking, and the images presented are in line with the quantification plots. Scale bars have been added.

6. *In figure-S4B, the co-localization of TLR9 and VAMP3 looks same before and after CpG-A stimulation in siBAD-LAMP cells in images. However, the statistical analyses indicate the increase of the co-localization by CpG-A stimulation. The authors need to explain this inconsistency or replace the results.*

The quantification represents the number of events per cell (n=50) and therefore give of a statistical relevance to our results. We realized that there was a duplication in the presented panels and a novel Figure S5B is now showing representative results for the iPLA.

7. *In all the statistical analyses, Pearson's coefficient analysis should show from 0 to 1 at Y axis.*

Given that significance in Pearson's coefficient calculation is only considered above 0.5, we feel that imaging quantification results are clearer and easier to read with the graduation starting from 0.4, which can be considered here as a zero to judge of co-localization and positive correlation significance.

8. *The length of scale bars and P values are often missing. They need to be described.*

All the scale bars and P values are homogeneous and were described in the material and methods.

Reviewer 2:

This manuscript reports on the role of the endo-lysosomal membrane protein BAD-LAMP and its impact on TLR9 trafficking, as well as downstream signaling events associated with it (cytokine production). In the absence of compelling biochemical data such as co-immunoprecipitation -the Pierre lab has state-of-the-art technology for these types of applications- claims of physical association cannot be sustained, as they are based solely on colocalization by immunofluorescence, a technique that lacks the resolution to claim direct interactions. The work is quite descriptive in nature, an obvious consequence of the technical approach chosen, and I doubt whether this work will influence thinking in the field, unless the fluorescence data were backed up by the molecular approach alluded to above. The proximity ligation assay samples too small a number of molecules (too few events) to allow an unambiguous interpretation of the proposed interactions.

We have to underline that most of the data presented here are performed with primary human pDCs and without using tagged-proteins (in particular TLR9-GFP) or transfections to follow their intracellular transport during time. Thus, this work could not be carried-out using other techniques than microscopy, and it is already a “tour de force” to present quantitative data to support our model. Although, descriptive for some parts, this work also provides many functional evidences on a novel role of BAD-LAMP in controlling signalling and cytokine production by regulating the trafficking of TLR9 during pDC activation. Our findings might not influence the “thinking in the field”, but they clarify the nature of the endosomes involved with TLR9 and IRF7 signalling, a point that has remained controversial in recent years, and they also underline the existence of an accessory molecule that is human specific and regulate endocytic TLRs transport, which is a completely novel finding.

Although following endogenous proteins in primary cells is one of the strength of this work, it renders, however, traditional biochemical approaches difficult to perform. Indeed, until now most of the biochemistry and cell biology performed on TLR9 and its interactors have been carried-out using overexpressed tagged proteins in non-relevant cell lines. Given that quality antibodies allowing a good human TLR9 biochemical analysis are not available, we would have to turn towards similar approaches to show that overexpressed and modified tagged-molecules can interact together in a unbalanced stoichiometric context. We feel that demonstrating a direct interaction of BAD-LAMP with TLR9 in these conditions will not change the conclusions of the paper and will not provide a strong enough support to our hypothesis, in absence of a complete structural and biophysical analysis of the interactions, which is clearly beyond the scope of this manuscript. We nevertheless present for this reviewer eyes the results, of our attempts to show that endogenous BAD-LAMP and TLR9 can interact in activated CAL-1 cells. Given that only a TLR9 proteolytically process fragment was revealed to interact with BAD-LAMP by co-IP, we feel that these data are encouraging and strongly suggest that the two molecules interact physically, however they are too preliminary to be included in the manuscript, given our current lack of tools to define the molecular nature of this TLR9 form.

Co-immunoprecipitation assay between BAD-LAMP and TLR9 on CAL-1 at steady state and upon CpG A treatment

CAL-1 cells stimulated for 0h to 6h with CpG were lysed in IP buffer and Immunoprecipitation was used using the 34.2 anti-Bad-LAMP antibody and Protein_A sepharose beads (n=2). Immunoprecipitated material was immunoblotted for TLR9 using anti-TLR9 (H-100, SC). This antibody detects mostly proteolytical fragment of TLR9 (INPUT), which were enriched in the immunoprecipitate of BAD-LAMP (IP anti-BAD), but not in the control IP with irrelevant antibodies (IP CT)

We were therefore careful of not to claim the existence of a physical association between TLR9 and BAD-LAMP in the "results" section of manuscript. However, given the resolution of the iPLA approach, we feel that this assay is sufficiently accurate and statistically relevant to put forward this hypothesis for discussion. Moreover, we have added novel experiments on the key role of the cytoplasmic tyrosine of BAD-LAMP in addressing TLR9 to late endosomes, further suggesting that a direct or indirect association of the two molecules is the most likely possibility to explain this effect.

Minor points: The text still contains many gallicisms that should be weeded out, regardless of where this work would be published (inappropriate use of plural nouns as modifiers of other nouns, no agreement between plural subject and singular verb, words like 'spacio-temporal' "specie-specific" etc., "we could not observe". The term intracellular FACS is a misnomer, as no sorting of any kind is performed on cells

We congratulate the reviewer for his/her knowledge of Latin etymology and have addressed in the text all his/her minor concerns.

Reviewer 3:

This is an impressive and technically challenging study with in-depth analysis at what is controlling TLR9 signaling in pDCs. The authors have used both primary cells or the cell line CAL-1 which although does not fully replicate the biology of pDC - in particular with low IFN response - allowed the authors to conduct more detailed mechanistic studies.

We thank this reviewer for his supportive opinion.

1- The authors observed that BAD-LAMP expression is reduced in pDCs following culture with CpG-A. This can be due to the presence of IFN induced by CpG-A, to TLR9 signaling or both. The authors show that IFN is enough to decrease BAD-LAMP but they should also show whether a CpG-B, which does not induce much IFN, can lead by itself to the same decrease and whether it then impacts the TLR9/MyD88/BAD complex formation.

In Figure S1, we have shown that anti-IFN blocking antibodies prevent BAD-LAMP down-modulation upon CpG-A stimulation suggesting that type-I IFN is the principle factor responsible for this phenomenon. As for other CpG ODNs, we have addressed this concern directly in the text, by referring to our previous paper describing BAD-LAMP expression in pDCs (Defays et al., Blood 2011), which showed that IL-3 and all CpG ODN forms can reduce BAD-LAMP expression, with the strongest effect observed with CpG-C ODN, while CpG-A and -B were relatively equivalent in their inhibitory activity. Clearly although CpG-B stimulation induces less type-I IFN than CpG-A, production still occurs in quantities that might be sufficient *in vitro* to trigger BAD-LAMP down-modulation, thus rendering the interpretation of the data difficult.

2- The authors are repeatedly raising the idea of a sequential involvement of the IRF and NF-kB pathways but are only using CpG-A in their experiments which are known to be poor activator the NF-kB pathway. Using a CpG-C would have seemed to be a better choice and the authors should explain the rationale behind their choice of reagents.

We agree with this reviewer, but CpG-C, by inducing all pathways at the same time, will have potentially complicated the precise dissection of the different events at work. Given the complexity to set up-the experiments and describe precisely the kinetics of these events, we used in priority CpG-A, which takes longer to diffuse along the entire endocytic pathway and allow a good time discrimination between the different signaling events occurring in the different endosomes. Importantly, CpG-A allowed-us to follow precisely BAD-LAMP trafficking and the recruitment of IRF7 by TLR9, in a clear and sequential manner, as well as the transient formation of the hybrid compartment in response to stimulation. In addition, CpG-A was also the ODN that gave us the best results to activate CAL-1 cells to produce type-I IFN, and therefore we decided to perform all our experiments using the same stimuli to be consistent in the different model systems (primary pDCs v.s. CAL-1 cells). We have amended our text to underline the possibility that sequential activation could be particularly observed with CpG-A

3- The data presented in figure 5 are key to the conclusions presented by the authors as the data show that the overexpression of BAD-LAMP in CAL-1 cells impacts TLR9 distribution and IFN/TNF production. As the level of BAD-LAMP is drastically increased (Sup fig 5) and the level constant for 6h, how do the authors explain the relatively modest impact on IFN and TNF. This should be discussed.

CAL-1 cells are not a match for primary pDCs in terms of type-I IFN production, and this is reflected by the relatively modest levels of induction presented in Fig 5C. We therefore work in a model system that has a relatively low dynamic range for cytokine production. Importantly by focusing on IRF-7 recruitment by TLR9 in the VAMP3⁺ compartment, we had to use CpG-A ODN, which is a strong type-I IFN inducer but poor NF-κB activator. Thus, the experiments were optimized to demonstrate the inhibitory effect of BAD-LAMP expression on type-I IFN production, which is quite severe, as shown in the Fig 5C. As for TNF-α, we observed a 10-fold increase in mRNA expression at 3 h, which is still a considerable for CpG-A-stimulated CAL-1 cells. Moreover, even if BAD-LAMP favors TLR9 access to late endosomes and increase the speed of activation, TLR9 degradation is likely to be increased over-time reducing the intensity of signaling and ultimately limiting BAD-LAMP influence over NF-κB signaling as displayed in the new immunoblot in Fig. 5 showing the p65 phosphorylation is enhanced before being strongly reduced after 6h upon BAD-LAMP ectopic expression.

4- It seems that the anti-TLR9 antibody used in the manuscript do not bind the TLR9 in all the organelles inside the pDCs or CAL-1 cells. One risk is that the altered staining could be due to non-specific binding. The authors should show a negative control using TLR9-negative cells.

We have provide for this reviewer's eye a confocal image of TLR9 staining in HEK293 cells expressing or not TLR9, to demonstrate the specificity of the anti-TLR9 antibodies used in this study. We did not feel however that it was necessary to integrate this data set in the manuscript. (see answer to reviewer 1).

5- In many of the figures, the authors are plotting IFN or TNF expression as fold changes. This can be misleading in particular as these 2 cytokines are not expressed at 0h. The authors should show the actual relative expression levels to the housekeeping gene.

It turns out that some extremely low levels of cytokines mRNA can be detected in non-treated cells and therefore this is impacting directly the measure of actual relative expression levels compared to the housekeeping genes. The chosen presentation is therefore the most adapted to not mislead the readers, since it truly reflects the variations observed comparatively among the different situations.

6- As a minor comment, the last paragraph of the introduction is not a place to repeat the abstract and should not be used to just summarize the main findings.

This was modified as requested

REVIEWERS' COMMENTS:

Reviewer #3 (Remarks to the Author):

The authors have improved the manuscript and have answered my concerns. I would appreciate however that the various points that are discussed by the authors in their rebuttal letter be incorporated in the manuscript. I think it will help readers better understand the rationale behind the use of certain reagents and will put the results in perspective.

NCOMMS-16-29671-T "BAD-LAMP controls TLR9 trafficking and signaling in human plasmacytoid dendritic cells"

Reviewer #3 (Remarks to the Author):

The authors have improved the manuscript and have answered my concerns. I would appreciate however that the various points that are discussed by the authors in their rebuttal letter be incorporated in the manuscript. I think it will help readers better understand the rationale behind the use of certain reagents and will put the results in perspective.

We have now added at p16 of the "Discussion" section, the following comments about the points discussed in the rebuttal letter, mostly focusing on co-immunoprecipitation and the use of iPLA.

"Until now, most of the biochemistry and cell biology performed on TLR9 and its interactors have been carried-out using overexpressed tagged-proteins (in particular TLR9-GFP) in non-relevant cell lines. Given that quality antibodies allowing a good human TLR9 biochemical analysis are not available, we had to turn to advanced microscopy techniques to evaluate the interactions of BAD-LAMP, UNC93B1 and TLR9 in primary human pDCs. iPLA was of great efficacy to demonstrate the close vicinity and likely interactions of these different endogenous molecules during time, however until now our attempts to demonstrate unequivocally a physical binding using traditional co-immunoprecipitation methods remained unsuccessful."